# Stochastic Optimization of Areas Under Precision-Recall Curves with Provable Convergence

**Qi Qi** [†][*], **Youzhi Luo**[‡][*], **Zhao Xu**[‡][*], **Shuiwang Ji**[‡], **Tianbao Yang**[†]

[†]Department of Computer Science, The University of Iowa
[‡]Department of Computer Science & Engineering, Texas A&M University
{qi-qi,tianbao-yang}@uiowa.edu, {yzluo,zhaoxu,sji}@tamu.edu

## Abstract

Areas under ROC (AUROC) and precision-recall curves (AUPRC) are common metrics for evaluating classification performance for imbalanced problems. Compared with AUROC, AUPRC is a more appropriate metric for highly imbalanced datasets. While stochastic optimization of AUROC has been studied extensively, principled stochastic optimization of AUPRC has been rarely explored. In this work, we propose a principled technical method to optimize AUPRC for deep learning. Our approach is based on maximizing the averaged precision (AP), which is an unbiased point estimator of AUPRC. We cast the objective into a sum of *coupled compositional functions* with inner functions dependent on random variables of the outer level. We propose efficient adaptive and non-adaptive stochastic algorithms named SOAP with *provable convergence guarantee under mild conditions* by leveraging recent advances in stochastic compositional optimization. Extensive experimental results on image and graph datasets demonstrate that our proposed method outperforms prior methods on imbalanced problems in terms of AUPRC. To the best of our knowledge, our work represents the first attempt to optimize AUPRC with provable convergence. The SOAP has been implemented in the libAUC library at `https://libauc.org/`.

## 1 Introduction

Although deep learning (DL) has achieved tremendous success in various domains, the standard DL methods have reached a plateau as the traditional objective functions in DL are no longer sufficient to model all requirements in new applications, which slows down the democratization of AI. For instance, in healthcare applications, data is often highly imbalanced, e.g., patients suffering from rare diseases are much less than those suffering from common diseases. In these applications, accuracy (the proportion of correctly predicted examples) is deemed as an inappropriate metric for evaluating the performance of a classifier. Instead, area under the curve (AUC), including area under ROC curve (AUROC) and area under the Precision-Recall curve (AUPRC), is widely used for assessing the performance of a model. However, optimizing accuracy on training data does not necessarily lead to a satisfactory solution to maximizing AUC [12].

To break the bottleneck for further advancement, DL must be empowered with the capability of efficiently handling novel objectives such as AUC. Recent studies have demonstrated great success along this direction by maximizing AUROC [60]. For example, Yuan et al. [60] proposed a robust deep AUROC maximization method with provable convergence and achieved great success for classification of medical image data. However, to the best of our knowledge, novel DL by maximizing AUPRC has not yet been studied thoroughly. Previous studies [14, 20] have found that when dealing

---

[*]Contribute Equally. Correspondence to qi-qi@uiowa.edu, tianbao-yang@uiowa.edu

35th Conference on Neural Information Processing Systems (NeurIPS 2021), virtual.

with highly skewed datasets, Precision-Recall (PR) curves could give a more informative picture of an algorithm's performance, which entails the development of efficient stochastic optimization algorithms for DL by maximizing AUPRC.

Compared with maximizing AUROC, maximizing AUPRC is more challenging. The challenges for optimization of AUPRC are two-fold. First, the analytical form of AUPRC by definition involves a complicated integral that is not readily estimated from model predictions of training examples. In practice, AUPRC is usually computed based on some point estimators, e.g., trapezoidal estimators and interpolation estimators of empirical curves, non-parametric average precision estimator, and parametric binomial estimator [3]. Among these estimators, non-parametric average precision (AP) is an unbiased estimate in the limit and can be directly computed based on the prediction scores of samples, which lends itself well to the task of model parameters optimization. Second, a surrogate function for AP is highly complicated and non-convex. In particular, an unbiased stochastic gradient is not readily computed, which makes existing stochastic algorithms such as SGD provide no convergence guarantee. Most existing works for maximizing AP-like function focus on how to compute an (approximate) gradient of the objective function [4, 6, 8, 11, 24, 38, 40, 43, 47, 48], which leave stochastic optimization of AP with provable convergence as an open question.

> *Can we design direct stochastic optimization algorithms both in SGD-style and Adam-style for maximizing AP with provable convergence guarantee?*

In this paper, we propose a systematic and principled solution for addressing this question towards maximizing AUPRC for DL. By using a surrogate loss in lieu of the indicator function in the definition of AP, we cast the objective into a sum of non-convex compositional functions, which resembles a two-level stochastic compositional optimization problem studied in the literature [52, 53]. However, different from existing two-level stochastic compositional functions, the inner functions in our problem are dependent on the random variable of the outer level, which requires us developing a tailored stochastic update for computing an error-controlled stochastic gradient estimator. Specifically, a key feature of the proposed method is to maintain and update two scalar quantities associated with each positive example for estimating the stochastic gradient of the individual precision score at the threshold specified by its prediction score. By leveraging recent advances in stochastic compositional optimization, we propose both adaptive (Adam-style) and non-adaptive (SGD-style) algorithms, and establish their convergence under mild conditions. We conduct comprehensive empirical studies on class imbalanced graph and image datasets for learning graph neural networks and deep convolutional neural networks, respectively. We demonstrate that the proposed method can consistently outperform prior approaches in terms of AUPRC. In addition, we show that our method achieves better results when the sample distribution is highly imbalanced between classes and is insensitive to mini-batch size.

## 2   Related Work

**AUROC Optimization.** AUROC optimization [2] has attracted significant attention in the literature. Recent success of DL by optimizing AUROC on large-scale medical image data has demonstrated the importance of large-scale stochastic optimization algorithms and the necessity of accurate surrogate function [60]. Earlier papers [25, 28] focus on learning a linear model based on the pairwise surrogate loss and could suffer from a high computational cost, which could be as high as quadratic of the size of training data. To address the computational challenge, online and stochastic optimization algorithms have been proposed [18, 35, 42, 58, 63]. Recently, [21, 22, 36, 57] proposed stochastic deep AUC maximization algorithms by formulating the problem as non-convex strongly-concave min-max optimization problem, and derived fast convergence rate under PL condition, and in federated learning setting as well [21]. More recently, Yuan et al. [60] demonstrated the success of their methods on medical image classification tasks, e.g., X-ray image classification, melanoma classification based on skin images. However, an algorithm that maximizes the AUROC might not necessarily maximize AUPRC, which entails the development of efficient algorithms for DL by maximizing AUPRC.

**AUPRC Optimization.** AUPRC optimization is much more challenging than AUROC optimization since the objective is even not decomposable over pairs of examples. Although AUPRC optimization has been considered in the literature (cf. [15, 47, 41] and references therein), efficient scalable algorithms for DL with provable convergence guarantee is still lacking. Some earlier works tackled

---

[2]In the literature, AUROC optimization is simply referred to as AUC optimization.

this problem by using traditional optimization techniques, e.g., hill climbing search [37], cutting-plane method [61], dynamic programming [50], and by developing acceleration techniques in the framework of SVM [39]. These approaches are not scalable to big data for DL. There is a long list of studies in information retrieval [5, 11, 38, 47] and computer vision [4, 6, 8, 9, 24, 40, 48, 43], which have made efforts towards maximizing the AP score. However, most of them focus on how to compute an approximate gradient of the AP function or its smooth approximation, and provide no convergence guarantee for stochastic optimization based on mini-batch averaging. Due to lack of principled design, these previous methods when applied to deep learning are sensitive to the mini-batch size [6, 47, 48] and usually require a large mini-batch size in order to achieve good performance. In contrast, our stochastic algorithms are designed in a principled way to guarantee convergence without requiring a large mini-batch size as confirmed by our studies as well. Recently, [15] formulates the objective function as a constrained optimization problem using a surrogate function, and then casts it into a min-max saddle-point problem, which facilitates the use of stochastic min-max algorithms. However, they do not provide any convergence analysis for AUPRC maximization. In contrast, this is the first work that directly optimizes a surrogate function of AP (an unbiased estimator of AUPRC in the limit) and provides theoretical convergence guarantee for the proposed stochastic algorithms.

**Stochastic Compositional Optimization.** Optimization of a two-level compositional function in the form of $\mathbb{E}_\xi[f(\mathbb{E}_\zeta[g(\mathbf{w}; \zeta)]; \xi)]$ where $\xi$ and $\zeta$ are independent random variables, or its finite-sum variant has been studied extensively in the literature [1, 10, 52, 27, 30, 31, 33, 34, 46, 53, 59, 62, 45]. In this paper, we formulate the surrogate function of AP into a similar but more complicated two-level compositional function of the form $\mathbb{E}_\xi[f(\mathbb{E}_\zeta g(\mathbf{w}; \zeta, \xi))]$, where $\xi$ and $\zeta$ are independent and $\xi$ has a finite support. The key difference between our formulated compositional function and the ones considered in previous work is that the inner function $g(\mathbf{w}; \zeta, \xi)$ also depends on the random variable $\xi$ of the outer level. Such subtle difference will complicate the algorithm design and the convergence analysis as well. Nevertheless, the proposed algorithm and its convergence analysis are built on previous studies of stochastic two-level compositional optimization.

# 3 The Proposed Method

**Notations.** We consider binary classification problem. Denote by $(\mathbf{x}, y)$ a data pair, where $\mathbf{x} \in \mathbb{R}^d$ denotes the input data and $y \in \{1, -1\}$ denotes its class label. Let $h(\mathbf{x}) = h_{\mathbf{w}}(\mathbf{x})$ denote the predictive function parameterized by a parameter vector $\mathbf{w} \in \mathbb{R}^D$ (e.g., a deep neural network). Denote by $\mathbf{I}(\cdot)$ an indicator function that outputs 1 if the argument is true and zero otherwise. To facilitate the presentation, denote by $X$ a random data, by $Y$ its label and by $F = h(X)$ its prediction score. Let $\mathcal{D} = \{(\mathbf{x}_1, y_1), \ldots, (\mathbf{x}_n, y_n)\}$ denote the set of all training examples and $\mathcal{D}_+ = \{\mathbf{x}_i : y_i = 1\}$ denote the set of all positive examples. Let $n_+ = |\mathcal{D}_+|$ denote the number of positive examples. $\mathbf{x}_i \sim \mathcal{D}$ means that $\mathbf{x}_i$ is randomly sampled from $\mathcal{D}$.

## 3.1 Background on AUPRC and its estimator AP

Following the work of Bamber [2], AUPRC is an average of the precision weighted by the probability of a given threshold, which can be expressed as

$$A = \int_{-\infty}^{\infty} \Pr(Y = 1 | F \geq c) d \Pr(F \leq c | Y = 1),$$

where $\Pr(Y = 1 | F \geq c)$ is the precision at the threshold value of $c$. The above integral is an importance-sampled Monte Carlo integral, by which we may interpret AUPRC as the fraction of positive examples among those examples whose output values exceed a randomly selected threshold $c \sim F(X) | Y = 1$.

For a finite set of examples $\mathcal{D} = \{(\mathbf{x}_i, y_i), i = 1, \ldots, n\}$ with the prediction score for each example $\mathbf{x}_i$ given by $h_{\mathbf{w}}(\mathbf{x}_i)$, we consider to use AP to approximate AUPRC, which is given by

$$\text{AP} = \frac{1}{n_+} \sum_{i=1}^{n} \mathbf{I}(y_i = 1) \frac{\sum_{s=1}^{n} \mathbf{I}(y_s = 1) \mathbf{I}(h_{\mathbf{w}}(\mathbf{x}_s) \geq h_{\mathbf{w}}(\mathbf{x}_i))}{\sum_{s=1}^{n} \mathbf{I}(h_{\mathbf{w}}(\mathbf{x}_s) \geq h_{\mathbf{w}}(\mathbf{x}_i))}, \quad (1)$$

where $n_+$ denotes the number of positive examples. It can be shown that AP is an unbiased estimator in the limit $n \to \infty$ [3].

However, the non-continuous indicator function $\mathbf{I}(h_{\mathbf{w}}(\mathbf{x}_s) \geq h_{\mathbf{w}}(\mathbf{x}_i))$ in both numerator and denominator in (1) makes the optimization non-tractable. To tackle this, we use a loss function $\ell(\mathbf{w}; \mathbf{x}_s, \mathbf{x}_i)$ as a surrogate function of $\mathbf{I}(h_{\mathbf{w}}(\mathbf{x}_s) \geq h_{\mathbf{w}}(\mathbf{x}_i))$. One can consider different surrogate losses, e.g., hinge loss, squared hinge loss, and smoothed hinge loss, and exponential loss. In this paper, we will consider a smooth surrogate loss function to facilitate the development of an optimization algorithm, e.g., a squared hinge loss $\ell(\mathbf{w}; \mathbf{x}_s; \mathbf{x}_i) = (\max\{m - (h_{\mathbf{w}}(\mathbf{x}_i) - h_{\mathbf{w}}(\mathbf{x}_s)), 0\})^2$, where $m$ is a margin parameter. Note that we do not require $\ell$ to be a convex function, hence one can also consider non-convex surrogate loss such as ramp loss. As a result, our problem becomes

$$\min_{\mathbf{w}} P(\mathbf{w}) = \frac{1}{n_+} \sum_{\mathbf{x}_i \in \mathcal{D}_+} \frac{- \sum_{s=1}^{n} \mathbf{I}(y_s = 1)\ell(\mathbf{w}; \mathbf{x}_s; \mathbf{x}_i)}{\sum_{s=1}^{n} \ell(\mathbf{w}; \mathbf{x}_s; \mathbf{x}_i)}. \tag{2}$$

## 3.2 Stochastic Optimization of AP (SOAP)

We cast the problem into a finite-sum of compositional functions. To this end, let us define a few notations:

$$g(\mathbf{w}; \mathbf{x}_j, \mathbf{x}_i) = [g_1(\mathbf{w}; \mathbf{x}_j, \mathbf{x}_i), g_2(\mathbf{w}; \mathbf{x}_j, \mathbf{x}_i)]^\top = [\ell(\mathbf{w}; \mathbf{x}_j, \mathbf{x}_i)\mathbf{I}(y_j = 1), \ell(\mathbf{w}; \mathbf{x}_j, \mathbf{x}_i)]^\top$$
$$g_{\mathbf{x}_i}(\mathbf{w}) = \mathbb{E}_{\mathbf{x}_j \sim \mathcal{D}}[g(\mathbf{w}; \mathbf{x}_j, \mathbf{x}_i)], \tag{3}$$

where $g_{\mathbf{x}_i}(\mathbf{w}) : \mathbb{R}^d \to \mathbb{R}^2$. Let $f(\mathbf{s}) = -\frac{s_1}{s_2} : \mathbb{R}^2 \to \mathbb{R}$. Then, we can write the objective function for maximizing AP as a sum of compositional functions:

$$P(\mathbf{w}) = \frac{1}{n_+} \sum_{\mathbf{x}_i \in \mathcal{D}_+} f(g_{\mathbf{x}_i}(\mathbf{w})) = \mathbb{E}_{\mathbf{x}_i \sim \mathcal{D}_+}[f(g_{\mathbf{x}_i}(\mathbf{w}))]. \tag{4}$$

We refer to the above problem as an instance of **two-level stochastic coupled compositional functions**. It is similar to the two-level stochastic compositional functions considered in literature [52, 53] but with a subtle difference. The difference is that in our formulation the inner function $g_{\mathbf{x}_i}(\mathbf{w}) = \mathbb{E}_{\mathbf{x}_j \sim \mathcal{D}}[g(\mathbf{w}; \mathbf{x}_j, \mathbf{x}_i)]$ depends on the random variable $\mathbf{x}_i$ of the outer level. This difference makes the proposed algorithm slightly complicated by estimating $g_{\mathbf{x}_i}(\mathbf{w})$ separately for each positive example. It also complicates the analysis of the proposed algorithms. Nevertheless, we can still employ the techniques developed for optimizing stochastic compositional functions to design the algorithms and develop the analysis for optimizing the objective (4).

In order to motivate the proposed method, let us consider how to compute the gradient of $P(\mathbf{w})$. Let the gradient of $g_{\mathbf{x}_i}(\mathbf{w})$ be denoted by $\nabla_{\mathbf{w}} g_{\mathbf{x}_i}(\mathbf{w})^\top = (\nabla_{\mathbf{w}}[g_{\mathbf{x}_i}(\mathbf{w})]_1, \nabla_{\mathbf{w}}[g_{\mathbf{x}_i}(\mathbf{w})]_2)$. Then we have

$$\nabla_{\mathbf{w}} P(\mathbf{w}) = \frac{1}{n_+} \sum_{\mathbf{x}_i \in \mathcal{D}_+} \nabla_{\mathbf{w}} g_{\mathbf{x}_i}(\mathbf{w})^\top \nabla f(g_{\mathbf{x}_i}(\mathbf{w}))$$
$$= \frac{1}{n_+} \sum_{\mathbf{x}_i \in \mathcal{D}_+} \nabla_{\mathbf{w}} g_{\mathbf{x}_i}(\mathbf{w})^\top \left( \frac{-1}{[g_{\mathbf{x}_i}(\mathbf{w})]_2}, \frac{[g_{\mathbf{x}_i}(\mathbf{w})]_1}{([g_{\mathbf{x}_i}(\mathbf{w})]_2)^2} \right)^\top. \tag{5}$$

The major cost for computing $\nabla_{\mathbf{w}} P(\mathbf{w})$ lies at evaluating $g_{\mathbf{x}_i}(\mathbf{w})$ and its gradient $\nabla_{\mathbf{w}} g_{\mathbf{x}_i}(\mathbf{w})$, which involves passing through all examples in $\mathcal{D}$.

To this end, we will approximate these quantities by stochastic samples. The gradient $\nabla_{\mathbf{w}} g_{\mathbf{x}_i}(\mathbf{w})$ can be simply approximated by the stochastic gradient, i.e.,

$$\widehat{\nabla}_{\mathbf{w}} g_{\mathbf{x}_i}(\mathbf{w}) = \begin{pmatrix} \frac{1}{B} \sum_{\mathbf{x}_j \in \mathcal{B}} \mathbf{I}(y_j = 1)\nabla\ell(\mathbf{w}; \mathbf{x}_j, \mathbf{x}_i) \\ \frac{1}{B} \sum_{\mathbf{x}_j \in \mathcal{B}} \nabla\ell(\mathbf{w}; \mathbf{x}_j, \mathbf{x}_i) \end{pmatrix}, \tag{6}$$

where $\mathcal{B}$ denote a set of $B$ random samples from $\mathcal{D}$. For estimating $g_{\mathbf{x}_i}(\mathbf{w}) = \mathbb{E}_{\mathbf{x}_j \sim \mathcal{D}} g(\mathbf{w}; \mathbf{x}_j, \mathbf{x}_i)$, however, we need to ensure its approximation error is controllable due to the compositional structure such that the convergence can be guaranteed. We borrow a technique from the literature of stochastic compositional optimization [52] by using moving average estimator for estimating $g_{\mathbf{x}_i}(\mathbf{w})$ for all positive examples. To this end, we will maintain a matrix $\mathbf{u} = [\mathbf{u}^1, \mathbf{u}^2]$ with each column indexable by any positive example, i.e., $\mathbf{u}_{\mathbf{x}_i}^1, \mathbf{u}_{\mathbf{x}_i}^2$ correspond to the moving average estimator of $[g_{\mathbf{x}_i}(\mathbf{w})]_1$ and $[g_{\mathbf{x}_i}(\mathbf{w})]_2$, respectively. The matrix $\mathbf{u}$ is updated by the subroutine UG in Algorithm 2, where $\gamma \in (0, 1)$ is a parameter. It is notable that in Step 3 of Algorithm 2, we clip the moving average update of $\mathbf{u}_{\mathbf{x}_i}^2$ by a lower bound $u_0$, which is a given parameter. This step can ensure the division in computing the stochastic gradient estimator in (7) always valid and is also important for convergence

**Algorithm 1:** SOAP

1: **Input:** $\gamma, \alpha, u_0$, and other parameters for SGD-stype update or Adam-stype update.
2: Initialize $\mathbf{w}_1 \in \mathbb{R}^d$, $\mathbf{u} \in \mathbb{R}^{|n_+| \times 2}$
3: **for** $t = 1, \ldots, T$ **do**
4:     Draw a batch of $B_+$ positive samples denoted by $\mathcal{B}_+$.
5:     Draw a batch of $B$ samples denoted by $\mathcal{B}$.
6:     $\mathbf{u} = \text{UG}(\mathcal{B}, \mathcal{B}_+, \mathbf{u}, \mathbf{w}_t, \gamma, u_0)$
7:     Compute (biased) Stochastic Gradient Estimator

$$G(\mathbf{w}_t) = \frac{1}{B_+} \sum_{\mathbf{x}_i \in \mathcal{B}_+} \sum_{\mathbf{x}_j \in \mathcal{B}} \frac{(\mathbf{u}_{\mathbf{x}_i}^1 - \mathbf{u}_{\mathbf{x}_i}^2 \mathbf{I}(\mathbf{y}_j = 1)) \nabla \ell(\mathbf{w}; \mathbf{x}_j, \mathbf{x}_i)}{B(\mathbf{u}_{\mathbf{x}_i}^2)^2} \tag{7}$$

8:     Update $\mathbf{w}_{t+1}$ by a SGD-style method or by a Adam-style method

$$\mathbf{w}_{t+1} = \text{UW}(\mathbf{w}_t, G(\mathbf{w}_t))$$

9: **end for**
10: **Return:** last solution.

analysis. With these stochastic estimators, we can compute an estimate of $\nabla P(\mathbf{w})$ by equation (7), where $\mathcal{B}_+$ includes a batch of sampled positive data. With this stochastic gradient estimator, we can employ SGD-style method and Adam-style shown in Algorithm 3 to update the model parameter $\mathbf{w}$. The final algorithm named as SOAP is presented in Algorithm 1.

**Algorithm 2:** $\text{UG}(\mathcal{B}, \mathcal{B}_+, \mathbf{u}, \mathbf{w}_t, \gamma, u_0)$

1: **for** each positive $\mathbf{x}_i \in \mathcal{B}_+$ **do**
2:     Compute

$$[\tilde{g}_{\mathbf{x}_i}(\mathbf{w}_t)]_1 = \frac{1}{|\mathcal{B}|} \sum_{\substack{x_j \in \mathcal{B} \\ y_j = 1}} \ell(\mathbf{w}_t; \mathbf{x}_j, \mathbf{x}_i)$$

$$[\tilde{g}_{\mathbf{x}_i}(\mathbf{w}_t)]_2 = \frac{1}{|\mathcal{B}|} \sum_{\mathbf{x}_j \in \mathcal{B}} \ell(\mathbf{w}_t; \mathbf{x}_j, \mathbf{x}_i)$$

3:     Compute
    $\mathbf{u}_{\mathbf{x}_i}^1 = (1 - \gamma)\mathbf{u}_{\mathbf{x}_i}^1 + \gamma[\tilde{g}_{\mathbf{x}_i}(\mathbf{w}_t)]_1$
    $\mathbf{u}_{\mathbf{x}_i}^2 = \max((1-\gamma)\mathbf{u}_{\mathbf{x}_i}^2 + \gamma[\tilde{g}_{\mathbf{x}_i}(\mathbf{w}_t)]_2, u_0)$
4: **end for**
5: **Return u**

**Algorithm 3:** $\text{UW}(\mathbf{w}_t, G(\mathbf{w}_t))$

1: Option 1: SGD-style update (paras: $\alpha$)
    $\mathbf{w}_{t+1} = \mathbf{w}_t - \alpha G(\mathbf{w}_t)$
2: Option 2: Adam-style update (paras: $\alpha, \epsilon, \eta_1, \eta_2$)
    $h_{t+1} = \eta_1 h_t + (1 - \eta_1)G(\mathbf{w}_t)$
    $v_{t+1} = \eta_2 \hat{v}_t + (1 - \eta_2)(G(\mathbf{w}_t))^2$
    $\mathbf{w}_{t+1} = \mathbf{w}_t - \alpha \dfrac{h_{t+1}}{\sqrt{\epsilon + \hat{v}_{t+1}}}$
    where $\hat{v}_t = v_t$ (Adam) or
    $\hat{v}_t = \max(\hat{v}_{t-1}, v_t)$ (AMSGrad)
3: **Return:** $\mathbf{w}_{t+1}$

### 3.3 Convergence Analysis

In this subsection, we present the convergence results of SOAP and also highlight its convergence analysis. To this end, we first present the following assumption.

**Assumption 1.** *Assume that (a) there exists $\Delta_1$ such that $P(\mathbf{w}_1) - \min_{\mathbf{w}} P(\mathbf{w}) \leq \Delta_1$; (b) there exist $C, M > 0$ such that $\ell(\mathbf{w}; \mathbf{x}_i, \mathbf{x}_i) \geq C$ for any $\mathbf{x}_i \in \mathcal{D}_+$, $\ell(\mathbf{w}; \mathbf{x}_j, \mathbf{x}_i) \leq M$, and $\ell(\mathbf{w}; \mathbf{x}_j, \mathbf{x}_i)$ is Lipscthiz continuous and smooth with respect to $\mathbf{w}$ for any $\mathbf{x}_i \in \mathcal{D}_+, \mathbf{x}_j \in \mathcal{D}$; (c) there exists $V > 0$ such that $\mathbb{E}_{\mathbf{x}_j \sim \mathcal{D}}[\|g(\mathbf{w}; \mathbf{x}_j, \mathbf{x}_i) - g_{\mathbf{x}_i}(\mathbf{w})\|^2] \leq V$, and $\mathbb{E}_{\mathbf{x}_j \sim \mathcal{D}}[\|\nabla g(\mathbf{w}; \mathbf{x}_j, \mathbf{x}_i) - \nabla g_{\mathbf{x}_i}(\mathbf{w})\|^2] \leq V$ for any $\mathbf{x}_i$.*

With a bounded score function $h_{\mathbf{w}}(\mathbf{x})$ the above assumption can be easily satisfied. Based on the above assumption, we can prove that the objective function $P(\mathbf{w})$ is smooth.

**Lemma 1.** *Suppose Assumption 1 holds, then there exists $L > 0$ such that $P(\cdot)$ is L-smooth. In addition, there exists $u_0 \geq C/n$ such that $g_{\mathbf{x}_i}(\mathbf{w}) \in \Omega = \{\mathbf{u} \in \mathbb{R}^2, 0 \leq [\mathbf{u}]_1 \leq M, u_0 \leq [\mathbf{u}]_2 \leq M\}, \forall \mathbf{x}_i \in \mathcal{D}_+$.*

Next, we highlight the convergence analysis of SOAP employing the SGD-stype update and include that for employing Adam-style update in the supplement. Without loss of generality, we assume $|\mathcal{B}_+| = 1$ and the positive sample in $\mathcal{B}_+$ is randomly selected from $\mathcal{D}_+$ with replacement. When the context is clear, we abuse the notations $g_i(\mathbf{w})$ and $\mathbf{u}_i$ to denote $g_{\mathbf{x}_i}(\mathbf{w})$ and $\mathbf{u}_{\mathbf{x}_i}$ below, respectively. We first establish the following lemma following the analysis of non-convex optimization.

**Lemma 2.** *With $\alpha \leq 1/2$, running $T$ iterations of SOAP (SGD-style) updates, we have*

$$\frac{\alpha}{2}\mathbb{E}[\sum_{t=1}^{T}\|\nabla P(\mathbf{w}_t)\|^2] \leq \mathbb{E}[\sum_{t}(P(\mathbf{w}_t) - P(\mathbf{w}_{t+1}))] + \frac{\alpha C_1}{2}\mathbb{E}[\sum_{t=1}^{T}\|g_{i_t}(\mathbf{w}_t) - \mathbf{u}_{i_t}\|^2] + \alpha^2 T C_2,$$

*where $i_t$ denotes the index of the sampled positive data at iteration $t$, $C_1$ and $C_2$ are proper constants.*

Our key contribution is the following lemma that bounds the second term in the above upper bound.

**Lemma 3.** *Suppose Assumption 1 holds, with $\mathbf{u}$ initialized by (6) for every $\mathbf{x}_i \in \mathcal{D}_+$ we have*

$$\mathbb{E}[\sum_{t=1}^{T}\|g_{i_t}(\mathbf{w}_t) - \mathbf{u}_{i_t}\|^2] \leq \frac{n_+ V}{\gamma} + \gamma V T + 2\frac{n_+^2 \alpha^2 T C_3}{\gamma^2}, \tag{8}$$

*where $C_3$ is a proper constant.*

**Remark:** The innovation of proving the above lemma is by grouping $\mathbf{u}_{i_t}, t = 1, \ldots, T$ into $n_+$ groups corresponding to the $n_+$ positive examples, and then establishing the recursion of the error $\|g_{i_t}(\mathbf{w}_t) - \mathbf{u}_{i_t}\|^2$ within each group, and then summing up these recursions together.

Based on the two lemmas above, we establish the following convergence of SOAP with a SGD-style update.

**Theorem 1.** *Suppose Assumption 1 holds, let the parameters be $\alpha = \frac{1}{n_+^{2/5} T^{3/5}}, \gamma = \frac{n_+^{2/5}}{T^{2/5}}, \forall\, t \in 1, \cdots, T$, and $T > n_+$. Then after running $T$ iterations, SOAP with a SGD-style update satisfies $\mathbb{E}\left[\frac{1}{T}\sum_{t=1}^{T}\|\nabla P(\mathbf{w}_t)\|^2\right] \leq O(\frac{n_+^{2/5}}{T^{2/5}})$, where $O$ suppresses constant numbers.*

**Remark:** To the best of our knowledge, this is the first time a stochastic algorithm was proved to converge for AP maximization.

Similarly, we can establish the following convergence of SOAP by employing an Adam-style update, specifically the AMSGrad update.

**Theorem 2.** *Suppose Assumption 1 holds, let the parameters $\eta_1 \leq \sqrt{\eta_2} \leq 1, \alpha = \frac{1}{n_+^{2/5} T^{3/5}}, \gamma = \frac{n_+^{2/5}}{T^{2/5}}, \forall\, t \in 1, \cdots, T$, and $T > n_+$. Then after running $T$ iterations, SOAP with an AMSGRAD update satisfies $\mathbb{E}\left[\frac{1}{T}\sum_{t=1}^{T}\|\nabla P(\mathbf{w}_t)\|^2\right] \leq O(\frac{n_+^{2/5}}{T^{2/5}})$, where $O$ suppresses constant numbers.*

## 4 Experiments

In this section, we evaluate the proposed method through comprehensive experiments on imbalanced datasets. We show that the proposed method can outperform prior state-of-the-art methods for imbalanced classification problems. In addition, we conduct experiments on (i) the effects of imbalance ratio; (ii) the insensitivity to batch size and (iii) the convergence speed on testing data; and observe that our method (i) is more advantageous when data is more imbalanced, (ii) is not sensitive to batch size, and (iii) converges faster than baseline methods.

Our proposed optimization algorithm is independent of specific datasets and tasks. Therefore, we perform experiments on both graph and image prediction tasks. In particular, the graph prediction tasks in the contexts of molecular property prediction and drug discovery suffer from very severe imbalance problems as positive labels are very rare while negative samples are abundantly available. Thus, we choose to use graph data intensively in our experiments. Additionally, the graph data we use allow us to vary the imbalance ratio to observe the performance change of different methods.

In all experiments, we compare our method with the following baseline methods. **CB-CE** refers to a method using a class-balanced weighed cross entropy loss function, in which the weights for positive and negative samples are adjusted with the strategy proposed by Cui et al. [13]. **Focal** is to up-weight the penalty on hard examples using focal loss [32]. **LDAM** refers to training with label-distribution-aware margin loss [7]. **AUC-M** is an AUROC maximization method using a surrogate loss [60]. In addition, we compare with three methods for optimizing AUPRC or AP, namely, the **MinMax** method [15] - a method for optimizing a discrete approximation of AUPRC, **SmoothAP** [4] - a method that optimizes a smoothed approximation of AP, and **FastAP** - a method that uses soft histogram binning to approximate the gradient of AP [6]. For all of these methods, we use the

Table 1: The test AUPRC on the image datasets with two ResNet models. We report the average AUPRC and standard deviation (within brackets) over 5 runs.

| Datasets | CIFAR-10 | | CIFAR-100 | |
|---|---|---|---|---|
| Networks | ResNet18 | ResNet34 | ResNet18 | ResNet34 |
| CE | 0.7155 ($\pm$ 0.0058) | 0.6844($\pm$ 0.0031) | 0.5946 ($\pm$ 0.0031) | 0.5792 ($\pm$ 0.0028) |
| CB-CE | 0.7325 ($\pm$ 0.0039) | 0.6936($\pm$0.0021) | 0.6165 ($\pm$ 0.0096) | 0.5632($\pm$ 0.0129) |
| Focal | 0.7183($\pm$ 0.0082) | 0.6943($\pm$ 0.0007) | 0.6107($\pm$ 0.0093) | 0.5585($\pm$ 0.0285) |
| LDAM | 0.7346 ($\pm$ 0.0125) | 0.6745($\pm$ 0.0043) | 0.6153 ($\pm$ 0.0100) | 0.5662($\pm$ 0.0212) |
| AUC-M | 0.7399($\pm$ 0.0013) | 0.6825($\pm$ 0.0089) | 0.6103 ($\pm$ 0.0075) | 0.5306($\pm$ 0.0230) |
| SmoothAP | 0.7365 ($\pm$ 0.0088) | 0.6909 ($\pm$ 0.0049) | 0.6071($\pm$ 0.0143) | 0.5208 ($\pm$ 0.0505) |
| FastAP | 0.7028 ($\pm$ 0.0341) | 0.6798 ($\pm$ 0.0032) | 0.5618($\pm$ 0.0351) | 0.5151($\pm$ 0.0450) |
| MinMax | 0.7228 ($\pm$ 0.0118) | 0.6806($\pm$ 0.0027) | 0.6071($\pm$ 0.0064) | 0.5518($\pm$ 0.0030) |
| SOAP | **0.7629**($\pm$ 0.0014) | **0.7012**($\pm$ 0.0056) | **0.6251** ($\pm$ 0.0053) | **0.6001**($\pm$ 0.0060) |

SGD-style with momentum optimization for image prediction tasks and the Adam-style optimization algorithms for graph prediction tasks and unless specified otherwise. We refer to **imbalance ratio** as the number of positive samples over the total number of examples of a considered set. The hyper-parameters of all methods are fine tuned using cross-validation with training/validation splits mentioned below. For AP maximization methods, we use a sigmoid function to produce the prediction score. For simplicity, we set $u_0 = 0$ for SOAP and encounter no numerical problems in experiments. As SOAP requires positive samples for updating **u** to approximate the gradient of surrogate objective, we use a data sampler which samples a few positive examples (e.g., 2) and some negative examples per iteration. The same sampler applies to all methods for fair comparison. The code for reproducing the results is released here [44].

## 4.1 Image Classification

**Data.** We first conduct experiments on three image datasets: CIFAR10, CIFAR100 and Melanoma dataset [49]. We construct imbalanced version of CIFAR10 and CIFAR100 for binary classification. In particular, for each dataset we manually take the last half of classes as positive class and first half of classes as negative class. To construct highly imbalanced data, we remove 98% of the positive images from the training data and keep the test data unchanged (i.e., the testing data is still balanced). And we split the training dataset into train/validation set at 80%/20% ratio. The Melanoma dataset is from a medical image Kaggle competition, which serves as a natural real imbalanced image dataset. It contains 33,126 labeled medical images, among which 584 images are related to malignant melanoma and labelled as positive samples. Since the test set used by Kaggle organization is not available, we manually split the training data into train/validation/test set at 80%/10%/10% ratio and report the achieved AUPRC on the test set by our method and baselines. The images of Melanoma dataset are always resized to have a resolution of $384 \times 384$ in our experiments.

**Setup.** We use two ResNet [23] models, *i.e.*, ResNet18 and ResNet34, as the backbone networks for image classification. For all methods except for CE, the ResNet models are initialized with a model pre-trained by CE with a SGD optimizer. We tune the learning rate in a range {1e-5, 1e-4, 1e-3, 1e-2} and the weight decay parameter in a range {1e-6, 1e-5, 1e-4}. Then the last fully connected layer is randomly re-initialized and the network is trained by different methods with the same weight decay parameter but other hyper-parameters individually tuned for fair comparison, e.g., we tune $\gamma$ of SOAP in a range {0.9, 0.99,0.999}, and tune $m$ in {0.5, 1, 2, 5, 10}. We refer to this scheme as two-stage training, which is widely used for imbalanced data [60]. We consistently observe that this strategy can bring the model to a good initialization state and improve the final performance of our method and baselines.

**Results.** Table 1 shows the AUPRC on testing sets of CIFAR-10 and CIFAR-100. We report the results on Melanoma in Table 3. We can observe that the proposed method SOAP outperforms all baselines. It is also striking to see that on Melanoma dataset, our proposed SOAP can outperform all baselines by a large margin, and all other methods have very poor performance. The reason is that the testing set of Melanoma is also imbalanced (imbalanced ratio=1.72%), while the testing sets of CIFAR-10 and CIFAR-100 are balanced. We also observe that the AUROC maximization (AUC-M) does not necessarily optimize AUPRC. We also plot the final PR curves in Figure 3 in the supplement.

Table 2: The test AUPRC values on the HIV and MUV datasets with three graph neural network models. We report the average AUPRC and standard deviation (within brackets) over 3 runs.

| Dataset | Method | GINE | MPNN | ML-MPNN |
|---|---|---|---|---|
| HIV | CE | 0.2774 ($\pm$ 0.0101) | 0.3197 ($\pm$ 0.0050) | 0.2988 ($\pm$ 0.0076) |
| | CB-CE | 0.3082 ($\pm$ 0.0101) | 0.3056 ($\pm$ 0.0018) | 0.3291 ($\pm$ 0.0189) |
| | Focal | 0.3179 ($\pm$ 0.0068) | 0.3136 ($\pm$ 0.0197) | 0.3279 ($\pm$ 0.0173) |
| | LDAM | 0.2904 ($\pm$ 0.0008) | 0.2994 ($\pm$ 0.0128) | 0.3044 ($\pm$ 0.0116) |
| | AUC-M | 0.2998 ($\pm$ 0.0010) | 0.2786 ($\pm$ 0.0456) | 0.3305 ($\pm$ 0.0165) |
| | SmothAP | 0.2686 ($\pm$ 0.0007) | 0.3276 ($\pm$ 0.0063) | 0.3235 ($\pm$ 0.0092) |
| | FastAP | 0.0169 ($\pm$ 0.0031) | 0.0826 ($\pm$ 0.0112) | 0.0202 ($\pm$ 0.0002) |
| | MinMax | 0.2874 ($\pm$ 0.0073) | 0.3119 ($\pm$ 0.0075) | 0.3098 ($\pm$ 0.0167) |
| | SOAP | **0.3385 ($\pm$ 0.0024)** | **0.3401 ($\pm$ 0.0045)** | **0.3547 ($\pm$ 0.0077)** |
| MUV | CE | 0.0017 ($\pm$0.0001) | 0.0021 ($\pm$0.0002) | 0.0025 ($\pm$0.0004) |
| | CB-CE | 0.0055 ($\pm$0.0011) | 0.0483 ($\pm$0.0083) | 0.0121 ($\pm$0.0016) |
| | Focal | 0.0041 ($\pm$0.0007) | 0.0281 ($\pm$0.0141) | 0.0122 ($\pm$0.0001) |
| | LDAM | 0.0044 ($\pm$0.0022) | 0.0118 ($\pm$0.0098) | 0.0059 ($\pm$0.0021) |
| | AUC-M | 0.0026 ($\pm$0.0001) | 0.0040 ($\pm$0.0012) | 0.0028 ($\pm$0.0012) |
| | SmoothAP | 0.0073 ($\pm$0.0012) | 0.0068 ($\pm$0.0038) | 0.0029 ($\pm$0.0005) |
| | FastAP | 0.0016 ($\pm$0.0000) | 0.0023 ($\pm$0.0021) | 0.0022 ($\pm$0.0012) |
| | MinMax | 0.0028 ($\pm$0.0008) | 0.0027 ($\pm$0.0005) | 0.0043 ($\pm$0.0015) |
| | SOAP | **0.0254 ($\pm$0.0261)** | **0.3352 ($\pm$0.0008)** | **0.0236 ($\pm$0.0038)** |

## 4.2 Graph Classification for Molecular Property Prediction

**Data.** To further demonstrate the advantages of our method, we conduct experiments on two graph classification datasets. We use the datasets HIV and MUV from the MoleculeNet [55], which is a benchmark for molecular property prediction. The HIV dataset has 41,913 molecules from the Drug Therapeutics Program (DTP), and the positive samples are molecules tested to have inhibition ability to HIV. The MUV dataset has 93,127 molecules from the PubChem library, and molecules are labelled by whether a bioassay property exists or not. Note that the MUV dataset provides labels of 17 properties in total and we only conduct experiments to predict the third property as this property is more imbalanced. The percentage of positive samples in HIV and MUV datasets are 3.51% and 0.20%, respectively. We use the split of train/validation/test set provided by MoleculeNet. Molecules are treated as 2D graphs in our experiments, and we use the feature extraction procedure of MoleculeKit [54] to obtain node features of graphs. The same data preprocessing is used for all of our experiments on graph data.

**Setup.** Many recent studies have shown that graph neural networks (GNNs) are powerful models for graph data analysis [29, 17, 16]. Hence, we use three different GNNs as the backbone network for graph classification, including the message passing neural network (**MPNN**) [19], an invariant of graph isomorphism network [56] named by **GINE** [26], and the multi-level message passing neural network (**ML-MPNN**) proposed by Wang et al. [54]. We use the same two-stage training scheme with a similar hyper-parameter tuning. We pre-train the networks by Adam with 100 epochs and a tuned initial learning rate 0.0005, which is decayed by half after 50 epochs.

**Results.** The achieved AUPRC on the test set by all methods are presented in Table 2. Results show that our method can outperform all baselines by a large margin in terms of AUPRC, regardless of which model structure is used. These results clearly demonstrate that our method is effective for classification problems in which the sample distribution is highly imbalanced between classes.

## 4.3 Graph Classification for Drug Discovery

**Data.** In addition to molecular property prediction, we explore applying our method to drug discovery. Recent studies have shown that GNNs are effective in drug discovery through predicting the antibacterial property of chemical compounds [51]. Such application scenarios involves training a GNN model on labeled datasets and making predictions on a large library of chemical compounds so as to discover new antibiotic. However, because the positive samples in the training data, *i.e.*, compounds known to have antibacterial property, are very rare, there exists very severe class imbalance.

We show that our method can serve as a useful solution to the above problem. We conduct experiments on the MIT AICURES dataset from an open challenge (`https://www.aicures.mit.edu/tasks`)

Table 3: The test AUPRC values on the MIT AICURES dataset with two graph neural networks, and on the Kaggle Melanoma dataset with two CNN models. We report the average AUPRC and standard deviation (within brackets) from 3 independent runs over 3 different train/validation/test splits.

| Data | MIT AICURES | | Kaggle Melanoma | |
|---|---|---|---|---|
| Networks | GINE | MPNN | ResNet18 | ResNet34 |
| CE | 0.5037 ($\pm$ 0.0718) | 0.6282 ($\pm$ 0.0634) | 0.0701 ($\pm$ 0.0031) | 0.0582 ($\pm$ 0.0016) |
| CB-CE | 0.5655 ($\pm$ 0.0453) | 0.6308 ($\pm$ 0.0263) | 0.0631 ($\pm$ 0.0065) | 0.0721 ($\pm$ 0.0054) |
| Focal | 0.5143 ($\pm$ 0.1062) | 0.5875 ($\pm$ 0.0774) | 0.0549 ($\pm$ 0.0083) | 0.0663 ($\pm$ 0.0034) |
| LDAM | 0.5236 ($\pm$ 0.0551) | 0.6489 ($\pm$ 0.0556) | 0.0547 ($\pm$ 0.0046) | 0.0539 ($\pm$ 0.0069) |
| AUC-M | 0.5149 ($\pm$ 0.0748) | 0.5542 ($\pm$ 0.0474) | 0.1013 ($\pm$ 0.0071) | 0.0972 ($\pm$ 0.0035) |
| SmothAP | 0.2899 ($\pm$ 0.0220) | 0.4081 ($\pm$ 0.0352) | 0.1981 ($\pm$ 0.0527) | 0.2787 ($\pm$ 0.0232) |
| FastAP | 0.4777 ($\pm$ 0.0896) | 0.4518 ($\pm$ 0.1495) | 0.0324 ($\pm$ 0.0087) | 0.0359 ($\pm$ 0.0062) |
| MinMax | 0.5292 ($\pm$ 0.0330) | 0.5774 ($\pm$ 0.0468) | 0.0593 ($\pm$ 0.0037) | 0.0663 ($\pm$ 0.0084) |
| SOAP | **0.6639 ($\pm$ 0.0515)** | **0.6547 ($\pm$ 0.0616)** | **0.2624 ($\pm$ 0.0410)** | **0.3152 ($\pm$ 0.0337)** |

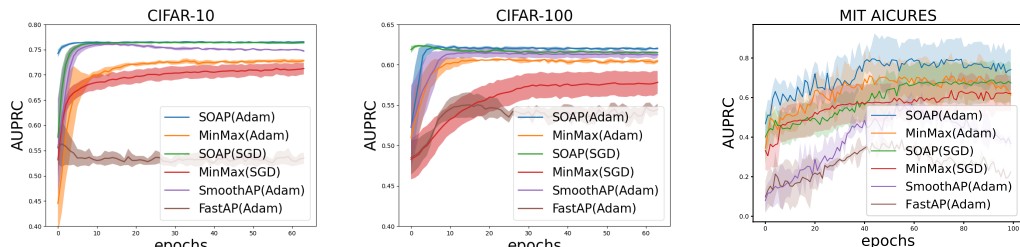

Figure 1: Comparison of convergence of different methods in terms of test AUPRC scores on CIFAR-10, CIFAR100 and MIT AICURES data.

in drug discovery. The dataset consists of 2097 molecules. There are 48 positive samples that have antibacterial activity to *Pseudomonas aeruginosa*, which is the pathogen leading to secondary lungs infections of COVID-19 patients. We conduct experiments on three random train/validation/test splits at 80%/10%/10% ratio, and report the average AUPRC on the test set over three splits.

**Setup.** Following the setup in Sec. 4.2, we use three GNNs: MPNN, GINE and ML-MPNN. We use the same two-stage training scheme with a similar hyper-parameter tuning. We pre-train GNNs by the Adam method for 100 epochs with a batch size of 64 and a tuned learning rate of 0.0005, which is decayed by half at the 50th epoch. Due to the limit of space, Table 3 only reports GINE and MPNN results. Please refer to Table 6 in the supplement for the full results of all three GNNs.

**Results.** The average test AUPRC from three independent runs over three splits are summarized in Table 3, Table 6. We can see that our SOAP can consistently outperform all baselines on all three GNN models. Our proposed optimization method can significantly improve the achieved AUPRC of GNN models, indicating that models tend to assign higher confidence scores to molecules with antibacterial activity. This can help identify a larger number of candidate drugs.

We have employed the proposed AUPRC maximization method for improving the testing performance on MIT AICures Challenge and achieved the 1st place. For details, please refer to [54].

### 4.4 Ablation Studies

**Effects of Imbalance Ratio.** We now study the effects of imbalance ratio on the performance improvements of our method. We use two datasets Tox21 and ToxCast from the MoleculeNet [55]. The Tox21 and ToxCast contain 8014 and 8589 molecules, respectively. There are 12 property prediction tasks in Tox21, and we conduct experiments on Task 0 and Task 2. Similarly, we select Task 12 and Task 8 of ToxCast for experiments. We use the split of train/validation/test set provided by MoleculeNet. The imbalanced ratios on the training sets are 4.14% for Task 0 of Tox21, 12.00% for Task 2 of Tox21, 2.97% for Task 12 of ToxCast, 8.67% for Task 8 of ToxCast.

Following Sec. 4.2, we test three neural network models MPNN, GINE and ML-MPNN. The hyper-parameters for training models are also the same as those in Sec. 4.2. We present the results of Tox21

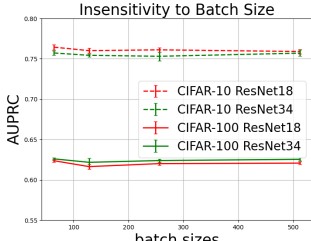
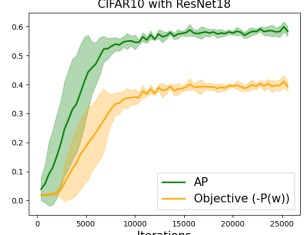
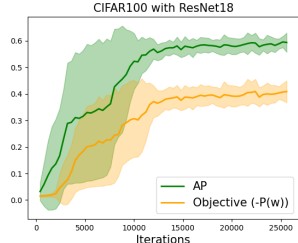

Figure 2: Left most: insensitivity to batch size of SOAP. Right two: consistency between AP and Surrogate Objective $-P(\mathbf{w})$ vs Iterations on CIFAR10 and CIFAR100.

Table 4: The test AUPRC over 3 independent runs by SOAP with different surrogate functions.

| Data | CIFAR10 | | CIFAR100 | |
|---|---|---|---|---|
| Networks | ResNet18 | ResNet34 | ResNet18 | ResNet34 |
| Squared Hinge | 0.7629 ($\pm$0.0014) | 0.7012 ($\pm$0.0056) | 0.6251 ($\pm$0.0053) | 0.6001 ($\pm$0.0060) |
| Logistic | 0.7542 ($\pm$0.0024) | 0.6968 ($\pm$0.0121) | 0.6378 ($\pm$0.0031) | 0.5923 ($\pm$0.0101) |
| Sigmoid | 0.7652 ($\pm$0.0035) | 0.6983 ($\pm$0.0084) | 0.6271 ($\pm$0.0043) | 0.5832 ($\pm$0.0054) |

| Data | HIV | | MUV | |
|---|---|---|---|---|
| Networks | GINE | MPNN | GINE | MPNN |
| Squared Hinge | 0.3485 ($\pm$0.0083) | 0.3401 ($\pm$0.0045) | 0.0354 ($\pm$0.0025) | 0.3365 ($\pm$0.0008) |
| Logistic | 0.3436 ($\pm$0.0043) | 0.3617 ($\pm$0.0031) | 0.0493 ($\pm$0.0261) | 0.3352 ($\pm$0.0008) |
| Sigmoid | 0.3387 ($\pm$0.0051) | 0.3629 ($\pm$0.0063) | 0.0298 ($\pm$0.0043) | 0.3362 ($\pm$0.0009) |

and ToxCast in Table 5 in the supplement. Our SOAP can consistently achieve improved performance when the data is extremely imbalanced. However, it sometimes fails to do so if the imbalance ratio is not too low. Clearly, the improvements from our method are higher when the imbalance ratio of labels is lower. In other words, our method is more advantageous for data with extreme class imbalance.

**Insensitivity to Batch Size.** We conduct experiments on CIFAR-10 and CIFAR-100 data by varying the mini-batch size for the SOAP algorithm and report results in Figure 2 (Left most). We can see that SOAP is not sensitive to the mini-batch size. This is consistent with our theory. In contrast, many previous methods for AP maximization are sensitive to the mini-batch size [47, 48, 6].

**Convergence Speed.** We report the convergence curves of different methods for maximizing AUPRC or AP in Figure 1 on different datasets. We can see that the proposed SOAP algorithms converge much faster than other baseline methods.

**More Surrogate Losses.** To verify the generality of SOAP, we evaluate the performance of SOAP with two more different surrogate loss functions $\ell(\mathbf{w}; \mathbf{x}_s, \mathbf{x}_i)$ as a surrogate function of the indicator $\mathbf{I}(h_\mathbf{w}(\mathbf{x}_s) \geq h_\mathbf{w}(\mathbf{x}_i))$, namely, the logistic loss, $\ell(\mathbf{w}; \mathbf{x}_s, \mathbf{x}_i) = -\log \frac{1}{1+\exp(-c(\ell(h_\mathbf{w}(\mathbf{x}_i)-h_\mathbf{w}(\mathbf{x}_s)))}$, and the sigmoid loss, $\ell(\mathbf{w}; \mathbf{x}_s, \mathbf{x}_i) = \frac{1}{1+\exp(c(\ell(h_\mathbf{w}(\mathbf{x}_i)-h_\mathbf{w}(\mathbf{x}_s)))}$ where $c$ is a hyperparameter. We tune $c \in \{1, 2\}$ in our experiments. We conduct experiments on CIFAR10, CIFAR100 following the experimental setting in Section 4.1 for the image data. For the graph data, we conduct experiments on HIV, MUV data following the experimental setting in Section 4.2. We report the results in Table 4. We can observe that SOAP has similar results with different surrogate loss functions.

**Consistency.** Finally, we show the consistency between the Surrogate Objective $-P(\mathbf{w})$ and AP by plotting the convergence curves on different datasets in Figure 2 (Right two). It is obvious two see the consistency between our surrogate objective and the true AP.

## 5 Conclusions and Outlook

In this work, we have proposed a stochastic method to optimize AUPRC that can be used in deep learning for tackling highly imbalanced data. Our approach is based on maximizing the averaged precision, and we cast the objective into a sum of coupled compositional functions. We proposed efficient adaptive and non-adaptive stochastic algorithms with provable convergence guarantee to compute the solutions. Extensive experimental results on graph and image datasets demonstrate that our proposed method can achieve promising results, especially when the class distribution is highly imbalanced. One limitation of SOAP is its convergence rate is still slow. In the future, we will consider to improve the convergence rate to address the limitation of the present work.

## Acknowledgments

We thank Bokun Wang for discussing the proofs, and thank anonymous reviewers for constructive comments. Q.Q contributed to the algorithm design, analysis, and experiments under supervision of T.Y. Y.L and Z.X contributed to the experiments under supervision of S.J. Q.Q and T.Y were partially supported by NSF Career Award #1844403, NSF Award #2110545 and NSF Award #1933212. Y.L, Z.X and S.J were partially supported by NSF IIS-1955189.

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
