# A Additional Experimental Results

We include the results about effect of imbalance ratio in Table 5, and the full results using three networks on MIT AICURES data in Table 6, and PR curves of final models on CIFAR10, CIFAR100 data in Figure 3.

Table 5: Test AUPRC on task 0 and task 2 of the Tox21 dataset and task 12 and task 8 of the ToxCast dataset with three graph neural network models.

| | | | |
|---|---|---|---|
| Tox21 Task 0 (Imbalance Ratio = 4.14%) | | | |
| Method | GINE | MPNN | ML-MPNN |
| CE | 0.4829 ($\pm$ 0.0123) | 0.5002 ($\pm$ 0.0054) | 0.4868 ($\pm$ 0.0048) |
| CB-CE | 0.4861 ($\pm$ 0.0113) | 0.4931 ($\pm$ 0.0068) | 0.4772 ($\pm$ 0.0033) |
| Focal | 0.4874 ($\pm$ 0.0148) | 0.4865 ($\pm$ 0.0067) | 0.4769 ($\pm$ 0.0134) |
| LDAM | 0.5093 ($\pm$ 0.0096) | 0.4823 ($\pm$ 0.0084) | 0.4709 ($\pm$ 0.0084) |
| AUC-M | 0.4356 ($\pm$ 0.0127) | 0.4428 ($\pm$ 0.0121) | 0.4632 ($\pm$ 0.0121) |
| SmoothAP | 0.3764 ($\pm$ 0.0053) | 0.4504 ($\pm$ 0.0089) | 0.4634 ($\pm$ 0.0064) |
| FastsAP | 0.0668 ($\pm$ 0.0061) | 0.2358 ($\pm$ 0.0093) | 0.0341 (0.0065) |
| MinMax (Adam) | 0.5066 ($\pm$ 0.0111) | 0.4940 ($\pm$ 0.0134) | 0.4947 ($\pm$ 0.0053) |
| SOAP (Adam) | **0.5276 ($\pm$ 0.0099)** | **0.5211 ($\pm$ 0.0089)** | **0.5093 ($\pm$ 0.0067)** |
| Tox21 Task 2 (Imbalance Ratio = 12.00%) | | | |
| Method | GINE | MPNN | ML-MPNN |
| CE | 0.5918 ($\pm$ 0.0063) | 0.6023 ($\pm$ 0.0087) | 0.5796 ($\pm$ 0.0071) |
| CB-CE | 0.5538 ($\pm$ 0.0087) | 0.5811 ($\pm$ 0.0095) | 0.5855 ($\pm$ 0.0069) |
| Focal | 0.5594 ($\pm$ 0.0069) | 0.6018 ($\pm$ 0.0083) | 0.5555 ($\pm$ 0.0025) |
| LDAM | 0.5369 ($\pm$ 0.0065) | 0.5991 ($\pm$ 0.0067) | 0.6014 ($\pm$ 0.0051) |
| AUC-M | 0.5832 ($\pm$ 0.0067) | 0.6117 ($\pm$ 0.0085) | 0.5987 ($\pm$ 0.0060) |
| SmoothAP | 0.5852 ($\pm$ 0.0045) | 0.6210 ($\pm$ 0.0069) | 0.4858 ($\pm$ 0.0061) |
| FastAP | 0.5605 ($\pm$ 0.0000) | 0.5605 ($\pm$ 0.0000) | 0.5605 ($\pm$ 0.0000) |
| MinMax (Adam) | 0.5623 ($\pm$ 0.0041) | 0.5977 ($\pm$ 0.0045) | 0.5079 ($\pm$ 0.0083) |
| SOAP (Adam) | **0.6172 ($\pm$ 0.0051)** | **0.6333 ($\pm$ 0.0160)** | **0.6196 ($\pm$ 0.0165)** |
| ToxCast Task 12 (Imbalance Ratio = 2.97%) | | | |
| Method | GINE | MPNN | ML-MPNN |
| CE | 0.0201 ($\pm$ 0.0031) | 0.0268 ($\pm$ 0.0031) | 0.0124 ($\pm$ 0.0031) |
| CB-CE | 0.0385 ($\pm$ 0.0042) | 0.0278 ($\pm$ 0.0073) | 0.0104 ($\pm$ 0.0029) |
| Focal | 0.0333 ($\pm$ 0.0052) | 0.0294 ($\pm$ 0.0043) | 0.0122 ($\pm$ 0.0024) |
| LDAM | 0.0217 ($\pm$ 0.0042) | 0.0298 ($\pm$ 0.0059) | 0.0179 ($\pm$ 0.0019) |
| AUC-M | 0.0333 ($\pm$ 0.0024) | 0.0454 ($\pm$ 0.0047) | 0.0089 ($\pm$ 0.0023) |
| SmoothAP | 0.227 ($\pm$ 0.0023) | 0.0208 ($\pm$ 0.0041) | 0.0079 ($\pm$ 0.0034) |
| FastAP | 0.0052 ($\pm$ 0.0048) | 0.0052 ($\pm$ 0.0038) | 0.0153 ($\pm$ 0.0013) |
| MinMax (Adam) | 0.0223 ($\pm$ 0.0033) | 0.0313 ($\pm$ 0.0061) | 0.0151 ($\pm$ 0.0023) |
| SOAP (Adam) | **0.0374 ($\pm$ 0.0025)** | **0.0601 ($\pm$ 0.0059)** | **0.0181 ($\pm$ 0.0023)** |
| ToxCast Task 8 (Imbalance Ratio = 8.67%) | | | |
| Method | GINE | MPNN | ML-MPNN |
| CE | 0.2071 ($\pm$ 0.0121) | 0.1101 ($\pm$ 0.0049) | 0.0923 ($\pm$ 0.0027) |
| CB-CE | 0.2089 ($\pm$ 0.0051) | 0.1349 ($\pm$ 0.0109) | 0.0734 ($\pm$ 0.0078) |
| Focal | 0.2011 ($\pm$ 0.0034) | 0.1223 ($\pm$ 0.0113) | 0.0792 ($\pm$ 0.0082) |
| LDAM | 0.1071 ($\pm$ 0.0101) | 0.1062 ($\pm$ 0.0104) | 0.0934 ($\pm$ 0.0125) |
| AUC-M | 0.0662 ($\pm$ 0.098) | 0.1258 ($\pm$ 0.0132) | 0.0979 ($\pm$ 0.0096) |
| SmoothAP | 0.0911 ($\pm$ 0.0123) | 0.1073 ($\pm$ 0.0011) | 0.0987 ($\pm$ 0.0049) |
| FastAP | 0.0999 ($\pm$ 0.0211) | 0.1037 ($\pm$ 0.0071) | 0.0932 ($\pm$ 0.0028) |
| MinMax (Adam) | 0.1381 ($\pm$ 0.0076) | 0.1173 ($\pm$ 0.0092) | 0.0903 ($\pm$ 0.0031) |
| SOAP (Adam) | **0.2561 ($\pm$ 0.0196)** | **0.1875 ($\pm$ 0.0124)** | **0.1107 ($\pm$ 0.0807)** |

Table 6: The test AUPRC values on the MIT AICURES dataset with three graph neural network models. We report the average AUPRC and standard deviation (within brackets) from 3 independent runs over 3 different train/validation/test splits.

| Method | GINE | MPNN | ML-MPNN |
|---|---|---|---|
| CE | 0.5037 ($\pm$ 0.0718) | 0.6282 ($\pm$ 0.0634) | 0.6101 ($\pm$ 0.1276) |
| CB-CE | 0.5655 ($\pm$ 0.0453) | 0.6308 ($\pm$ 0.0263) | 0.4903 ($\pm$ 0.1507) |
| Focal | 0.5143 ($\pm$ 0.1062) | 0.5875 ($\pm$ 0.0774) | 0.4718 ($\pm$ 0.0691) |
| LDAM | 0.5236 ($\pm$ 0.0551) | 0.6489 ($\pm$ 0.0556) | 0.6725 ($\pm$ 0.0594) |
| AUC-M | 0.5149 ($\pm$ 0.0748) | 0.5542 ($\pm$ 0.0474) | 0.4429 ($\pm$ 0.0486) |
| SmothAP | 0.2899 ($\pm$ 0.0220) | 0.4081 ($\pm$ 0.0352) | 0.4212 ($\pm$ 0.0507) |
| FastAP | 0.4777 ($\pm$ 0.0896) | 0.4518 ($\pm$ 0.1495) | 0.5174 ($\pm$ 0.0150) |
| MinMax | 0.5292 ($\pm$ 0.0330) | 0.5774 ($\pm$ 0.0468) | 0.5832 ($\pm$ 0.1080) |
| SOAP | **0.6639 ($\pm$ 0.0515)** | **0.6547 ($\pm$ 0.0616)** | **0.6503 ($\pm$ 0.0532)** |

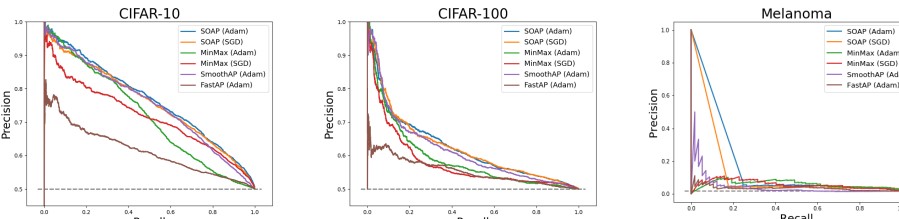

Figure 3: Precision-Recall curves of different methods on test dataset of CIFAR10, CIFAR100 and Melanoma datasets. The gray dashed lines are the random classifiers on test data sets whose AUPRC equals to the ratio between positive samples and all samples $n_+/n$ on every data set, respectively.

# B   Analysis of SOAP (SGD-style)

In the following, we abuse the notations $g_i(\mathbf{w}) = g_{\mathbf{x}_i}(\mathbf{w}) \in \mathbb{R}^2$ and $\mathbf{u}_i = \mathbf{u}_{\mathbf{x}_i} = ([\mathbf{u}_{\mathbf{x}_i}]_1, [\mathbf{u}_{\mathbf{x}_i}]_2)$. We use $\mathbf{u}_{i_t}$ to denote the updated vector at the $t$-th iteration for the sampled $i_t$-th positive data.

### B.1   Proof of Theorem 1

*Proof.* By combining Lemma 3 and Lemma 2, we have:

$$\frac{\alpha}{2}\mathbb{E}[\sum_{t=1}^T \|\nabla P(\mathbf{w}_t)\|^2] \le \mathbb{E}[\sum_t (P(\mathbf{w}_t) - P(\mathbf{w}_{t+1}))] + \frac{\alpha C_1}{2}\mathbb{E}[\sum_{t=1}^T \|g_{i_t}(\mathbf{w}_t) - \mathbf{u}_{i_t}\|^2] + \alpha^2 T C_2$$

$$\le \mathbb{E}[\sum_t (P(\mathbf{w}_t) - P(\mathbf{w}_{t+1}))] + \frac{\alpha C_1}{2}\Big\{\frac{n_+ V}{\gamma} + 2\gamma VT + 2\frac{n_+^2 \alpha^2 T C_3}{\gamma^2}\Big\} + \alpha^2 T C_2$$

$$\le \mathbb{E}_t[P(\mathbf{w}_1)] - \mathbb{E}_t[P(\mathbf{w}_{t+1})] + \frac{\alpha C_1}{2}\Big\{\frac{n_+ V}{\gamma} + 2\gamma VT + 2\frac{n_+^2 \alpha^2 T C_3}{\gamma^2}\Big\} + \alpha^2 T C_2$$

Then by set $\alpha = \frac{1}{n_+^{2/5} T^{3/5}}$, $\gamma = \frac{n_+^{2/5}}{T^{2/5}}$, and multiply $\frac{2}{\alpha T}$ on both sides of above equation,

$$\frac{1}{T}\mathbb{E}[\sum_{t=1}^T \|\nabla P(\mathbf{w}_t)\|^2] \le \frac{2\Delta_1}{T\alpha} + C_1\Big\{\frac{n_+ V}{\gamma T} + 2\gamma V + 2\frac{n_+^2 \alpha^2 C_3}{\gamma^2}\Big\} + \alpha C_2$$

$$\le \frac{2\Delta_1 n_+^{2/5}}{T^{2/5}} + C_1\Big\{\frac{n_+^{3/5} V}{T^{3/5}} + 2\frac{n_+^{2/5}}{T^{2/5}} + 2\frac{n_+^{2/5} C_3}{T^{2/5}}\Big\} + \frac{C_2}{n_+^{2/5} T^{3/5}}$$

$$\le O(\frac{n_+^{2/5}}{T^{2/5}})$$

where the last inequality is due to $T \ge n_+$ and $O$ compresses constant numbers. We finish the proof. $\square$

## B.2 Proof of Lemma 1

*Proof of Lemma 1.* We first prove the second part that $g_i(\mathbf{w}) \in \Omega$. Due to the definition of $g_i(\mathbf{w}) = \mathbb{E}_{\mathbf{x}_j \sim \mathcal{D}}[g(\mathbf{w}; \mathbf{x}_j, \mathbf{x}_i)] = \mathbb{E}_{\mathbf{x}_j \sim \mathcal{D}}[\ell(\mathbf{w}; \mathbf{x}_j, \mathbf{x}_i)\mathbf{I}(y_j = 1), \ell(\mathbf{w}; \mathbf{x}_i, \mathbf{x}_j)]$, and the Assumption 1, it is obvious to see that $0 \leq [g_i(\mathbf{w})]_1 \leq M$ and $M \geq [g_i(\mathbf{w})]_2 \geq C/n$ for all $i$, i.e., $g_i(\mathbf{w}) \in \Omega$. Next, we prove the smoothness of $P(\mathbf{w})$. To this end, we need to use the following Lemma 4 and the proof will be presented after Lemma 1.

**Lemma 4.** *Let* $L_f = 4(u_0 + M)/u_0^3, C_f = (u_0 + M)/u_0^2, L_g = \sqrt{2}L_l, C_g = \sqrt{2}C_l$, *then* $f(\mathbf{u})$ *is a* $L_f$ *-smooth,* $C_f$-*Lipschitz continuous function for any* $\mathbf{u} \in \Omega$, *and* $\forall \, i \in [1, \cdots n]$, $g_i$ *is a* $L_g$-*smooth,* $C_g$-*Lipschitz continuous function.*

Since $P(\mathbf{w}) = \frac{1}{n_+} \sum_{\mathbf{x}_i \in \mathcal{D}_+} f(g_i(\mathbf{w}))$. We first show $P_i(\mathbf{w}) = f(g_i(\mathbf{w}))$ is smooth. To see this,
$$\|\nabla P_i(\mathbf{w}) - \nabla P_i(\mathbf{w}')\| = \|\nabla g_i(\mathbf{w})^\top \nabla f(g_i(\mathbf{w})) - \nabla g_i(\mathbf{w}')^\top \nabla f(g_i(\mathbf{w}'))\|$$
$$\leq \|\nabla g_i(\mathbf{w})^\top \nabla f(g_i(\mathbf{w})) - \nabla g_i(\mathbf{w}')^\top \nabla f(g_i(\mathbf{w}))\|$$
$$+ \|\nabla g_i(\mathbf{w}')^\top \nabla f(g_i(\mathbf{w})) - \nabla g_i(\mathbf{w}')^\top \nabla f(g_i(\mathbf{w}'))\|$$
$$\leq C_f L_g \|\mathbf{w} - \mathbf{w}'\| + C_g L_f C_g \|\mathbf{w} - \mathbf{w}'\| = (C_f L_g + L_f C_g^2)\|\mathbf{w} - \mathbf{w}'\|.$$
Hence $P(\mathbf{w})$ is also $L = (C_f L_g + L_f C_g^2)$-smooth.

$\square$

## B.3 Proof of Lemma 4

*Proof of Lemma 4.* According to the definition, we have
$$f(\mathbf{u}) = \frac{-[\mathbf{u}]_1}{[\mathbf{u}]_2} \quad, \nabla_\mathbf{u} f(\mathbf{u}) = \left( \frac{-1}{[\mathbf{u}]_2}, \frac{[\mathbf{u}]_1}{([\mathbf{u}]_2)^2} \right)^\top, \quad \nabla_\mathbf{u}^2 f(\mathbf{u}) = \begin{pmatrix} 0, \frac{1}{([\mathbf{u}]_2)^2} \\ \frac{1}{([\mathbf{u}]_2)^2}, -\frac{2[\mathbf{u}]_1}{([\mathbf{u}]_2)^3} \end{pmatrix} \qquad (9)$$
Due to the assumption that $\ell(\mathbf{w}; \mathbf{x}_j, \mathbf{x}_i)$ is a $L_l$-smooth, $C_l$-Lipschitz continuous function, we have

$$\|\nabla_\mathbf{w} g_i(\mathbf{w})\|^2 \leq 2\|\frac{1}{n}\sum_{j=1}^n \nabla \ell_\mathbf{w}(\mathbf{w}; \mathbf{x}_j, \mathbf{x}_i)\|^2 \leq 2C_l^2 = C_g^2$$

$$\|\nabla_\mathbf{w} g_i(\mathbf{w}) - \nabla_\mathbf{w} g(\mathbf{w}')\|^2 \leq \|\frac{1}{n}\sum_{j=1}^n \nabla_\mathbf{w} \ell(\mathbf{w}; \mathbf{x}_j, \mathbf{x}_i) - \frac{1}{n}\sum_{j=1}^n \nabla_\mathbf{w} \ell(\mathbf{w}; \mathbf{x}_j, \mathbf{x}_i)\|^2$$

$$+ \|\frac{1}{n}\sum_{j=1}^n \nabla_\mathbf{w} \ell(\mathbf{w}; \mathbf{x}_j, \mathbf{x}_i)\mathbf{I}(y_j = 1) - \frac{1}{n}\sum_{j=1}^n \nabla_\mathbf{w} \ell(\mathbf{w}; \mathbf{x}_j, \mathbf{x}_i)\mathbf{I}(y_j = 1)\|^2 \leq 2L_l^2 = L_g^2 \quad (10)$$

$$\|\nabla f(\mathbf{u})\| \leq \sqrt{\frac{1}{[\mathbf{u}]_2^2} + \frac{[\mathbf{u}]_1^2}{[\mathbf{u}]_2^4}} \leq \frac{u_0 + M}{u_0^2} = C_f$$

$$\|\nabla^2 f(\mathbf{u})\| \leq \sqrt{\frac{2}{[\mathbf{u}]_2^4} + 4\frac{[\mathbf{u}]_1^2}{[\mathbf{u}]_2^6}} \leq \frac{4(u_0 + M)}{u_0^3} = L_f$$
We finish the proof of Lemma 4. $\square$

## B.4 Proof of Lemma 2

*Proof of Lemma 2.* To make the proof clear, we write $\nabla g_{i_t}(\mathbf{w}; \xi) = \nabla g(\mathbf{w}_t; \xi, \mathbf{x}_{i_t}), \xi \sim \mathcal{D}$. Let $\mathbf{u}_{i_t}$ denote the updated $\mathbf{u}$ vector at the $t$-th iteration for the selected positive data $i_t$.
$$P(\mathbf{w}_{t+1}) - P(\mathbf{w}_t) \leq \nabla P(\mathbf{w}_t)^\top (\mathbf{w}_{t+1} - \mathbf{w}_t) + \frac{L}{2}\|\mathbf{w}_{t+1} - \mathbf{w}_t\|^2$$

$$= -\alpha\|\nabla P(\mathbf{w}_t)\|^2 + \alpha \nabla P(\mathbf{w}_t)^\top (\nabla P(\mathbf{w}_t) - \nabla g_{i_t}^\top(\mathbf{w}_t; \xi)\nabla f(\mathbf{u}_{i_t})) + \frac{\alpha^2\|G(\mathbf{w}_t)\|^2 L}{2}$$

$$\leq -\alpha\|\nabla P(\mathbf{w}_t)\|^2 + \alpha \nabla P(\mathbf{w}_t)^\top (\nabla P(\mathbf{w}_t) - \nabla g_{i_t}^\top(\mathbf{w}_t; \xi)\nabla f(\mathbf{u}_{i_t})) + \alpha^2 C_2$$
where $C_2 = \|G(\mathbf{w}_t)\|^2 L/2 \leq C_g^2 C_f^2 L/2$.

Taking expectation on both sides, we have

$$\mathbb{E}_t[P(\mathbf{w}_{t+1})] \leq \mathbb{E}_t[P(\mathbf{w}_t) + \nabla P(\mathbf{w}_t)^\top (\mathbf{w}_{t+1} - \mathbf{w}_t) + \frac{L}{2}\|\mathbf{w}_{t+1} - \mathbf{w}_t\|^2]$$

$$= \mathbb{E}_t[P(\mathbf{w}_t) - \alpha\|\nabla P(\mathbf{w}_t)\|^2 + \alpha\nabla P(\mathbf{w}_t)^\top(\nabla P(\mathbf{w}_t) - \nabla g_{i_t}(\mathbf{w}_t;\xi)^\top \nabla f(\mathbf{u}_{i_t})) + \alpha^2 C_2$$

$$= P(\mathbf{w}_t) - \alpha\|\nabla P(\mathbf{w}_t)\|^2 + \alpha\nabla P(\mathbf{w}_t)^\top(\mathbb{E}_t[\nabla P(\mathbf{w}_t) - \nabla g_{i_t}(\mathbf{w}_t;\xi)^\top \nabla f(\mathbf{u}_{i_t})]) + \alpha^2 C_2$$

where $\mathbb{E}_t$ means taking expectation over $i_t, \xi$ given $\mathbf{w}_t$.
Noting that $\nabla P(\mathbf{w}_t) = \mathbb{E}_{i_t,\xi}[\nabla g_{i_t}(\mathbf{w}_t;\xi)^\top \nabla f(g_{i_t}(\mathbf{w}_t))]$, where $i_t$ and $\xi$ are independent.

$$\mathbb{E}_t[P(\mathbf{w}_{t+1})] - P(\mathbf{w}_t)$$

$$\leq -\alpha\|\nabla P(\mathbf{w}_t)\|^2 + \alpha\nabla P(\mathbf{w}_t)^\top(\mathbb{E}_t[\nabla g_{i_t}(\mathbf{w}_t;\xi)^\top \nabla f(g_{i_t}(\mathbf{w}_t))] - \mathbb{E}_t[\nabla g_{i_t}(\mathbf{w}_t;\xi)^\top \nabla f(\mathbf{u}_{i_t})]) + \alpha^2 C_2$$

$$= -\alpha\|\nabla P(\mathbf{w}_t)\|^2 + \mathbb{E}_t[\alpha\nabla P(\mathbf{w}_t)^\top(\nabla g_{i_t}(\mathbf{w}_t;\xi)^\top \nabla f(g_{i_t}(\mathbf{w}_t)) - \nabla g_{i_t}(\mathbf{w}_t;\xi)^\top \nabla f(\mathbf{u}_{i_t}))] + \alpha^2 C_2$$

$$\overset{(a)}{\leq} -\alpha\|\nabla P(\mathbf{w}_t)\|^2 + \mathbb{E}_t[\frac{\alpha}{2}\|\nabla P(\mathbf{w}_t)\|^2 + \frac{\alpha}{2}\|\nabla g_{i_t}(\mathbf{w}_t;\xi)^\top \nabla f(g_{i_t}(\mathbf{w}_t)) - \nabla g_{i_t}(\mathbf{w}_t;\xi)^\top \nabla f(\mathbf{u}_{i_t})\|^2 + \alpha^2 C_2$$

$$\overset{(b)}{\leq} -\alpha\|\nabla P(\mathbf{w}_t)\|^2 + \mathbb{E}_t[\frac{\alpha}{2}\|\nabla P(\mathbf{w}_t)\|^2 + \frac{\alpha C_1}{2}\|g_{i_t}(\mathbf{w}_t) - \mathbf{u}_{i_t}\|^2 + \alpha^2 C_2$$

$$= -(\alpha - \frac{\alpha}{2})\|\nabla P(\mathbf{w}_t)\|^2 + \frac{\alpha C_1}{2}\mathbb{E}_t[\|g_{i_t}(\mathbf{w}_t) - \mathbf{u}_{i_t}\|^2] + \alpha^2 C_2$$

where the equality (a) is due to $ab \leq a^2/2 + b^2/2$ and the inequality $(b)$ uses the factor $\|\nabla g_{i_t}(\mathbf{w}_t;\xi)\| \leq C_l$ and $\nabla f$ is $L_f$-Lipschitz continuous for $\mathbf{u}, \mathbf{g}_i(\mathbf{w}) \in \Omega$ and $C_1 = C_l^2 C_f^2$. Hence we have,

$$\frac{\alpha}{2}\|\nabla P(\mathbf{w}_t)\|^2 \leq P(\mathbf{w}_t) - \mathbb{E}_t[P(\mathbf{w}_{t+1})] + \frac{\alpha C_1}{2}\mathbb{E}_t[\|g_{i_t}(\mathbf{w}_t) - \mathbf{u}_{i_t}\|^2] + \alpha^2 C_2$$

Taking summation and expectation over all randomness, we have

$$\frac{\alpha}{2}\mathbb{E}[\sum_{t=1}^T \|\nabla P(\mathbf{w}_t)\|^2] \leq \mathbb{E}[\sum_t (P(\mathbf{w}_t) - P(\mathbf{w}_{t+1}))] + \frac{\alpha C_1}{2}\mathbb{E}[\sum_{t=1}^T \|g_{i_t}(\mathbf{w}_t) - \mathbf{u}_{i_t}\|^2] + \alpha^2 C_2 T$$

□

## B.5 Proof of Lemma 3

Let $i_t$ denote the selected positive data $i_t$ at $t$-th iteration. We will divide $\{1, \ldots, T\}$ into $n_+$ groups with the $i$-th group given by $\mathcal{T}_i = \{t_1^i, \ldots, t_k^i \ldots, \}$, where $t_k^i$ denotes the iteration that the $i$-th positive data is selected at the $k$-th time for updating $\mathbf{u}$. Let us define $\phi(t) : [T] \to [n_+] \times [T]$ that maps the selected data into its group index and within group index, i.e, there is an one-to-one correspondence between index $t$ and selected data $i$ and its index within $\mathcal{T}_i$. Below, we use notations $a_i^k$ to denote $a_{t_k^i}$. Let $T_i = |\mathcal{T}_i|$. Hence, $\sum_{i=1}^{n_+} T_i = T$.

*Proof of Lemma 3.* To prove Lemma 3, we first introduce another lemma that establishes a recursion for $\|\mathbf{u}_{i_t} - g_{i_t}(\mathbf{w}_t)\|^2$, whose proof is presented later.

**Lemma 5.** *By the updates of SOAP Adam-style or SGD-style with $\mathcal{B}_+ = 1$, the following equation holds for $\forall\, t \in 1, \cdots, T$*

$$\mathbb{E}_t[\|\mathbf{u}_{i_t} - g_{i_t}(\mathbf{w}_t)\|^2] \overset{\phi(t)}{=} \mathbb{E}_t[\|\mathbf{u}_i^k - g_i(\mathbf{w}_i^k)\|^2]$$
$$\leq (1-\gamma)\|\mathbf{u}_i^{k-1} - g_i(\mathbf{w}_i^{k-1})\|^2 + \gamma^2 V + \gamma^{-1}\alpha^2 n_+^2 C_3 \tag{11}$$

*where $\mathbb{E}_t$ denotes the conditional expectation conditioned on history before $t_{k-1}^i$.*

Then, by mapping every $i_t$ to its own group and make use of Lemma 5, we have

$$\mathbb{E}[\sum_{k=0}^{K_i} \|\mathbf{u}_i^k - g_i^k(\mathbf{w}_i^k)\|^2] \leq \mathbb{E}\left[\frac{[\|\mathbf{u}_i^0 - g_i(\mathbf{w}_i^0)\|^2]}{\gamma} + \gamma V T_i + \gamma^{-2} n_+^2 C_3 \alpha^2 T_i\right] \tag{12}$$

where $\mathbf{u}_i^0$ is the initial vector for $\mathbf{u}_i$, which can be computed by a mini-batch averaging estimator of $g_i(\mathbf{w}_0)$. Thus

$$\mathbb{E}[\sum_{t=1}^{T}\|g_{i_t}(\mathbf{w}_t) - \mathbf{u}_{i_t}\|^2] \overset{\phi(t)}{=} \mathbb{E}[\sum_{i=1}^{n_+}\sum_{k=0}^{K_i}\|\mathbf{u}_i^k - g_i^k(\mathbf{w}_i^k)\|^2]$$

$$\leq \sum_{i=1}^{n_+}\left\{\frac{[\|\mathbf{u}_i^0 - g_i^0(\mathbf{w}_i^0)\|^2]}{\gamma} + \gamma V\mathbb{E}[T_i] + \gamma^{-2}n_+^2 C_3\alpha^2\mathbb{E}[T_i]\right\}$$

$$\leq \frac{n_+ V}{\gamma} + \gamma VT + \frac{n_+^2\alpha^2 TC_3}{\gamma^2}$$

$\square$

## B.6 Proof of Lemma 5

*Proof.* We first introduce the following lemma, whose proof is presented later.

**Lemma 6.** *Suppose the sequence generated in the training process using the positive sample $i$ is $\{\mathbf{w}_{i_1}^i, \mathbf{w}_{i_2}^i, \ldots, \mathbf{w}_{i_{T_i}}^i\}$, where $0 < i_1 < i_2 < \cdots < i_{T_i} \leq T$, then $\mathbb{E}_{|i_k}[i_{k+1} - i_k] \leq n_+, and, \mathbb{E}_{|i_k}[(i_{k+1} - i_k)^2] \leq 2n_+^2, \forall k.$*

Define $\widetilde{g}_{i_t}(\mathbf{w}_t) = g(\mathbf{w}_t, \xi, \mathbf{x}_{i_t})$. Let $\prod_\Omega(\cdot) : \mathbb{R}^2 \to \Omega$ denotes the projection operator. By the updates of $\mathbf{u}_{i_t}$, we have $\mathbf{u}_{i_t} = \mathbf{u}_i^k = \prod_\Omega[(1 - \gamma)\mathbf{u}_i^{k-1} + \gamma\widetilde{g}_{i_t}(\mathbf{w}_t)]$.

$$\mathbb{E}_t[\|\mathbf{u}_{i_t} - g_{i_t}(\mathbf{w}_t)\|^2] \overset{\phi(t)}{=} \mathbb{E}[\|\mathbf{u}_i^k - g_i(\mathbf{w}_i^k)\|^2]$$

$$= \mathbb{E}_t[\|\prod_\Omega((1-\gamma)\mathbf{u}_i^{k-1} + \gamma\widetilde{g}_i(\mathbf{w}_i^k)) - \prod_\Omega(g_i(\mathbf{w}_t))\|^2]$$

$$\leq \mathbb{E}_t[\|((1-\gamma)\mathbf{u}_i^{k-1} + \gamma\widetilde{g}_i(\mathbf{w}_i^k) - g_i(\mathbf{w}_t)\|^2]$$

$$\leq \mathbb{E}_t[\|((1-\gamma)(\mathbf{u}_i^{k-1} - g_i(\mathbf{w}_i^{k-1})) + \gamma(\widetilde{g}_i(\mathbf{w}_i^k) - g_i(\mathbf{w}_i^k)) + (1-\gamma)(g_i(\mathbf{w}_i^{k-1}) - g_i(\mathbf{w}_i^k))\|^2]$$

$$\leq \mathbb{E}_t[\|((1-\gamma)(\mathbf{u}_i^{k-1} - g_i(\mathbf{w}_i^{k-1})) + (1-\gamma)(g_i(\mathbf{w}_i^{k-1}) - g_i(\mathbf{w}_i^k))\|^2] + \gamma^2 V$$

$$\leq [(1-\gamma)^2(1+\gamma)\|\mathbf{u}_i^{k-1} - g_i(\mathbf{w}_i^{k-1})\|^2] + \gamma^2 V + \frac{(1+\gamma)(1-\gamma)^2}{\gamma}C_g\mathbb{E}[\|\mathbf{w}_i^k - \mathbf{w}_i^{k-1}\|^2]$$

$$\leq [(1-\gamma)\|\mathbf{u}_i^{k-1} - g_i(\mathbf{w}_i^{k-1})\|^2] + \gamma^2 V + \gamma^{-1}\alpha^2 C_g\mathbb{E}_t[\|\sum_{t=t_{k-1}^i}^{t_k^i-1}\nabla g_{i_t}(\mathbf{w}_t;\xi)\nabla f(\mathbf{u}_{i_t})\|^2]$$

$$\leq [(1-\gamma)\|\mathbf{u}_i^{k-1} - g_i(\mathbf{w}_i^{k-1})\|^2] + \gamma^2 V + \gamma^{-1}\alpha^2 C_g\mathbb{E}_t[(t_k^i - t_{k-1}^i)^2]C_g^2 C_f^2)]$$

$$\overset{(a)}{\leq} \mathbb{E}[(1-\gamma)\|\mathbf{u}_i^{k-1} - g_i(\mathbf{w}_i^{k-1})\|^2] + \gamma^2 V + 2\gamma^{-1}\alpha^2 n_+^2 C_g^3 C_f^2$$

$$\leq [(1-\gamma)\|\mathbf{u}_i^{k-1} - g_i(\mathbf{w}_i^{k-1})\|^2] + \gamma^2 V + \gamma^{-1}\alpha^2 n_+^2 C_3$$

where the inequality (a) is due to that $t_k^i - t_{k-1}^i$ is a geometric distribution random variable with $p = 1/n_+$, i.e., $\mathbb{E}_{|t_{k-1}^i}[(t_k^i - t_{k-1}^i)^2] \leq 2/p^2 = 2n_+^2$, by Lemma 6. The last equality hold by defining $C_3 = 2C_g^3 C_f^2$.

$\square$

## B.7 Proof of Lemma 6

*Proof.* Proof of Lemma 6. Denote the random variable $\Delta_k = i_{k+1} - i_k$ that represents the iterations that the $i$th positive sample has been randomly selected for the $k + 1$-th time conditioned on $i_k$. Then $\Delta_k$ follows a Geometric distribution such that $\Pr(\Delta_k = j) = (1 - p)^{j-1}p$, where $p = \frac{1}{n_+}$, $j = 1, 2, 3, \cdots$. As a result, $\mathbb{E}[\Delta_k|i_k] = 1/p = n_+$. $\mathbb{E}[\Delta_k^2|i_k] = \mathrm{Var}(\Delta_k) + \mathbb{E}[\Delta_k|i_k]^2 = \frac{1-p}{p^2} + \frac{1}{p^2} \leq \frac{2}{p^2} = 2n_+^2$. $\square$

## C    Proof of Theorem 2 (SOAP with Adam-Style Update)

*Proof.* We first provide two useful lemmas, whose proof are presented later.

**Lemma 7.** *Assume assumption 1 holds*
$$\|\mathbf{w}_{t+1} - \mathbf{w}_t\|^2 \leq \alpha^2 d(1 - \eta_2)^{-1}(1 - \tau)^{-1} \tag{13}$$
*where $d$ is the dimension of $\mathbf{w}$, $\eta_1 < \sqrt{\eta_2} < 1$, and $\tau := \eta_1^2/\eta_2$.*

**Lemma 8.** *With $c = (1 + (1 - \eta_1)^{-1})\epsilon^{-\frac{1}{2}} C_g^2 L_f^2$, running $T$ iterations of SOAP (Adam-style) updates, we have*

$$\sum_{t=1}^{T} \frac{\alpha(1 - \eta_1)(\epsilon + C_g^2 C_f^2)^{-1/2}}{2} \|\nabla P(\mathbf{w}_t)\|^2 \leq \mathbb{E}[\mathcal{L}_1] - \mathbb{E}[\mathcal{L}_{T+1}]$$

$$+ 2\eta_1 L\alpha^2 Td(1 - \eta_1)^{-1}(1 - \eta_2)^{-1}(1 - \tau)^{-1} + L\alpha^2 Td(1 - \eta_2)^{-1}(1 - \tau)^{-1} \tag{14}$$

$$+ 2(1 - \eta_1)^{-1}\alpha C_g^2 C_f^2 \sum_{i'=1}^{d}((\epsilon + \hat{v}_0^{i'})^{-1/2}) + c\alpha \sum_{t=1}^{T} \mathbb{E}_t[\|g_{i_t}(\mathbf{w}_t) - \mathbf{u}_{i_t}\|^2]$$

*where $\mathcal{L}_{t+1} = P(\mathbf{w}_{t+1}) - c_{t+1}\langle \nabla P(\mathbf{w}_t), D_{t+1}h_{t+1}\rangle$.*

According to Lemma 8 and plugging Lemma 3 into equation (14), we have

$$\sum_{t=1}^{T} \frac{\alpha(1 - \eta_1)(\epsilon + C_g^2 C_f^2)^{-1/2}}{2} \|\nabla P(\mathbf{w}_t)\|^2$$

$$\leq \mathbb{E}[\mathcal{L}_1] - \mathbb{E}[\mathcal{L}_{T+1}] + 2\eta_1 L\alpha^2 Td(1 - \eta_1)^{-1}(1 - \eta_2)^{-1}(1 - \tau)^{-1} + L\alpha^2 dT(1 - \eta_2)^{-1}(1 - \tau)^{-1}$$

$$+ 2c\alpha C_g^2 C_f^2 \sum_{i'=1}^{d}(\epsilon + \hat{v}_0^{i'})^{-1/2} + c\alpha(\frac{n_+ V}{\gamma} + 2\gamma VT + \frac{2C_g n_+^2 C_3 \alpha^2 T}{\gamma^2}) \tag{15}$$

Let $\eta' = (1 - \eta_2)^{-1}(1 - \tau)^{-1}, \eta'' = (1 - \eta_1)^{-1}(1 - \eta_2)^{-1}(1 - \tau)^{-1}$, and $\widetilde{\eta} = (1 - \eta_1)^{-2}(1 - \eta_2)^{-1}(1 - \tau)^{-1}$. As $(1 - \eta_1)^{-1} \geq 1, (1 - \eta_2)^{-1} \geq 1$, then $\widetilde{\eta} \geq \eta'' \geq \eta' \geq 1$.

Then by rearranging terms in Equation (15), dividing $\alpha T(1 + \eta_1)(\epsilon + C_g^2 C_f^2)^{-1/2}$ on both sides and suppress constants, $C_g, L_g, C_3, L, C_f, L_f, V, \epsilon$ into big $O$, we get

$$\frac{1}{T}\sum_{t=1}^{T} \|\nabla P(\mathbf{w}_t)\|^2 \leq \frac{1}{\alpha T(1 - \eta_1)} O\Big(\mathbb{E}[\mathcal{L}_1] - \mathbb{E}[\mathcal{L}_{T+1}] + \eta''\eta_1\alpha^2 Td + \eta'\alpha^2 Td + \alpha\sum_{i'=1}^{d}(\epsilon + \hat{v}_0^{i'})^{-1/2}$$

$$+ \frac{c\alpha n_+}{\gamma} + c\alpha\gamma T + \frac{c\alpha^3 n_+^2 T}{\gamma^2}\Big)$$

$$\overset{(a)}{\leq} \frac{1}{\alpha T(1 - \eta_1)} O\Big(\mathbb{E}[\mathcal{L}_1] - \mathbb{E}[\mathcal{L}_{T+1}] + \eta''\eta_1\alpha^2 Td + \eta'\alpha^2 Td + \alpha d(\epsilon + C_f C_g)^{-1/2}$$

$$+ \frac{c\alpha n_+}{\gamma} + c\alpha\gamma T + \frac{c\alpha^3 n_+^2 T}{\gamma^2}\Big)$$

$$\overset{(b)}{\leq} \frac{\widetilde{\eta}}{\alpha T} O\Big(\mathbb{E}[\mathcal{L}_1] - [\mathcal{L}_{T+1}] + (1 + \eta_1)\alpha^2 Td + \alpha d + \frac{c\alpha n_+}{\gamma} + c\alpha\gamma T + \frac{c\alpha^3 n_+^2 T}{\gamma^2}\Big) \tag{16}$$

where the inequality $(a)$ is due to $\hat{v}_0^{i'} = G^{i'}(\mathbf{w}_0)^2 \leq \|G(\mathbf{w}_0)\|^2 \leq C_f^2 C_g^2$. The last inequality $(b)$ is due to $\widetilde{\eta} \geq \eta'' \geq \eta' \geq 1$.

Moreover, by the definition of $\mathcal{L}$ and $\mathbf{w}_0 = \mathbf{w}_1$, we have

$$\mathbb{E}[\mathcal{L}_1] = P(\mathbf{w}_1) - c_1\langle\nabla P(\mathbf{w}_0), D_1 h_1\rangle \le P(\mathbf{w}_1) + c_1\|\nabla P(\mathbf{w}_0)\|\|\mathbf{w}_1 - \mathbf{w}_0\|\frac{1}{\alpha} = P(\mathbf{w}_1)$$

$$-\mathbb{E}[\mathcal{L}_{T+1}] \le -P(\mathbf{w}_{T+1}) + c_{T+1}\langle\nabla P(\mathbf{w}_T), D_T h_T\rangle$$

$$\le -\min_{\mathbf{w}} P(\mathbf{w}) + c_{T+1}\|\nabla P(\mathbf{w}_{t-1})\|\|\mathbf{w}_{t+1} - \mathbf{w}_t\|\frac{1}{\alpha}$$

$$\overset{(a)}{\le} -\min_{\mathbf{w}} P(\mathbf{w}) + (1 - \eta_1)^{-1}\alpha\sqrt{d}(1 - \eta_2)^{-1/2}(1 - \tau)^{-1/2}$$

$$\overset{(b)}{\le} -\min_{\mathbf{w}} P(\mathbf{w}) + \widetilde{\eta}\sqrt{d}\alpha$$

(17)

where the inequality $(a)$ is due to Lemma 7 and $c_{T+1} \le (1 - \eta_1)^{-1}\alpha$ in equation (30). The inequality $(b)$ is due to $(1 - \eta_1)^{-1}(1 - \eta_2)^{-1/2}(1 - \tau)^{-1/2} \le (1 - \eta_1)^{-1}(1 - \eta_2)^{-1}(1 - \tau)^{-1} \le \eta'' \le \widetilde{\eta}$. Thus $\mathbb{E}[\mathcal{L}_1] - \mathbb{E}[\mathcal{L}_{T+1}] \le P(\mathbf{w}_1) - \min_{\mathbf{w}} P(\mathbf{w}) + \widetilde{\eta}\sqrt{d}\alpha \le \Delta_1 + \widetilde{\eta}\sqrt{d}\alpha$ by combining equation (16) and (17).

Then we have

$$\frac{1}{T}\sum_{t=1}^{T}\|\nabla P(\mathbf{w}_t)\|^2 \le \widetilde{\eta}O\Big(\frac{\Delta_1 + \widetilde{\eta}\sqrt{d}\alpha}{\alpha T} + (1 + \eta_1)\alpha d + \frac{d}{T} + \frac{n_+ c}{T\gamma} + c\gamma + \frac{\alpha^2 n_+^2}{\gamma^2}\Big)$$

$$\overset{(a)}{\le} \widetilde{\eta}O\Big(\frac{\Delta_1 n_+^{2/5}}{T^{2/5}} + \frac{\widetilde{\eta}\sqrt{d}}{T} + \frac{(1 + \eta_1)d}{n_+^{2/5}T^{3/5}} + \frac{d}{T} + \frac{cn_+^{3/5}}{T^{3/5}} + 2\frac{cn_+^{2/5}}{T^{2/5}}\Big)$$

(18)

$$\overset{(b)}{\le} O\big(\frac{n_+^{2/5}}{T^{2/5}}\big)$$

The inequality $(a)$ is due to $\gamma = \frac{n_+^{2/5}}{T^{2/5}}$, $\alpha = \frac{1}{n_+^{2/5}T^{3/5}}$. In inequality $(b)$, we further compress the $\Delta_1$, $\eta_1, \widetilde{\eta}, c$ into big $O$ and $\gamma \le 1 \to n_+^{2/5} \le T^{2/5}$.

$\square$

## C.1  Proof of Lemma 7

*Proof.* This proof is following the proof of Lemma 4 in [10].

Choosing $\eta_1 < 1$ and defining $\tau = \frac{\eta_1^2}{\eta_2}$, with the Adam-style (Algorithm 3) updates of SOAP that $h_{t+1} = \eta_1 h_t + (1 - \eta_1)G(\mathbf{w}_t)$, we can verify for every dimension $l$,

$$|h_{t+1}^l| = |\eta_1 h_t^l + (1 - \eta_1)G^l(\mathbf{w}_t)| \le \eta_1|h_t^l| + |G^l(\mathbf{w}_t)|$$

$$\le \eta_1(\eta_1|h_{t-1}^l| + |G^l(\mathbf{w}_{t-1})|) + |G^l(\mathbf{w}_t)|$$

$$\le \sum_{p=0}^{t}\eta_1^{t-p}|G^l(\mathbf{w}_p)| = \sum_{p=0}^{t}\sqrt{\tau}^{t-p}\sqrt{\eta_2}^{t-p}|G^l(\mathbf{w}_p)|$$

$$\le \Big(\sum_{p=0}^{t}\tau^{t-p}\Big)^{\frac{1}{2}}\Big(\sum_{p=0}^{t}\eta_2^{t-p}(G^l(\mathbf{w}_p))^2\Big)^{\frac{1}{2}}$$

(19)

$$\le (1 - \tau)^{-\frac{1}{2}}\Big(\sum_{p=0}^{t}\eta_2^{t-p}(G^l(\mathbf{w}_t))^2\Big)^{\frac{1}{2}}$$

where $\mathbf{w}^l$ is the $l$th dimension of $\mathbf{w}$, the third inequality follows the Cauchy-Schwartz inequality. For the $l$th dimension of $\hat{v}$, $\hat{v}_t^l$, first we have $\hat{v}_1^l \ge (1 - \eta_2)(G^l(\mathbf{w}_1)^2)$. Then since

$$\hat{v}_{t+1}^l \ge \eta_t\hat{v}_t^l + (1 - \eta_2)(G^l(\mathbf{w}_t))^2$$

by induction we have

$$\hat{v}_{t+1}^l \ge (1 - \eta_2)\sum_{p=0}^{t}\eta_2^{t-p}(G^l(\mathbf{w}_t))^2$$

(20)

Using equation (19) and equation (20), we have

$$|h_{t+1}^l|^2 \le (1-\tau)^{-1}\Big(\sum_{p=0}^{t} \eta_2^{t-p}(G^l(\mathbf{w}_t))^2\Big) \tag{21}$$

$$\le (1-\eta_2)^{-1}(1-\tau)^{-1}\hat{v}_{t+1}^l$$

Then follow the Adam-style update in Algorithm 3, we have

$$\|\mathbf{w}_{t+1} - \mathbf{w}_t\|^2 = \alpha^2 \sum_{l=1}^{d}(\epsilon + \hat{v}_{t+1}^l)^{-1}|h_{t+1}^l|^2 \le \alpha^2 d(1-\eta_2)^{-1}(1-\tau)^{-1} \tag{22}$$

which completes the proof. □

## C.2  Proof of Lemma 8

*Proof.* To make the proof clear, we make some definitions the same as the proof of Lemma 2. Denote by $\nabla g_{i_t}(\mathbf{w}_t; \xi) = \nabla g(\mathbf{w}_t; \xi, \mathbf{x}_{i_t}), \xi \sim \mathcal{D}$, where $i_t$ is a positive sample randomly generated from $\mathcal{D}_+$ at $t$-th iteration, and $\xi$ is a random sample that generated from $\mathcal{D}$ at $t$-th iteration. It is worth to notice that $i_t$ and $\xi$ are independent. $\mathbf{u}_{i_t}$ denote the updated $\mathbf{u}$ vector at the $t$-th iteration for the selected positive data $i_t$.

$$P(\mathbf{w}_{t+1}) \le P(\mathbf{w}_t) + \nabla P(\mathbf{w}_t)^\top(\mathbf{w}_{t+1} - \mathbf{w}_t) + \frac{L}{2}\|\mathbf{w}_{t+1} - \mathbf{w}_t\|^2$$

$$\le P(\mathbf{w}_t) - \alpha \nabla P(\mathbf{w}_t)^\top(D_{t+1}h_{t+1}) + \alpha^2 d(1-\eta_2)^{-1}(1-\tau)^{-1}L/2$$

where $D_{t+1} = \frac{1}{\sqrt{\epsilon I + \hat{\mathbf{v}}_{t+1}}}$, $h_{t+1} = \eta_1 h_t + (1-\eta_1)\nabla g_{i_t}^\top(\mathbf{w}_t; \xi)\nabla f(\mathbf{u}_{i_t})$ and the second inequality is due to Lemma 7. Taking expectation on both sides, we have

$$\mathbb{E}_t[P(\mathbf{w}_{t+1})] \le P(\mathbf{w}_t) \underbrace{-\mathbb{E}_t[\nabla P(\mathbf{w}_t)^\top(D_{t+1}h_{t+1})]}_{\Upsilon}\alpha + \alpha^2 d(1-\eta_2)^{-1}(1-\tau)^{-1}L$$

where $\mathbb{E}_t[\cdot] = \mathbb{E}[\cdot|\mathcal{F}_t]$ implies taking expectation over $i_t, \xi$ given $\mathbf{w}_t$. In the following analysis, we decompose $\Upsilon$ into three parts and bound them one by one:

$$\Upsilon = -\langle\nabla P(\mathbf{w}_t), D_{t+1}h_{t+1}\rangle = -\langle\nabla P(\mathbf{w}_t), D_t h_{t+1}\rangle - \langle\nabla P(\mathbf{w}_t), (D_{t+1}-D_t)h_{t+1}\rangle$$

$$= -(1-\eta_1)\langle\nabla P(\mathbf{w}_t), D_t\nabla g_{i_t}(\mathbf{w}_t; \xi)^\top\nabla f(\mathbf{u}_{i_t})\rangle - \eta_1\langle\nabla P(\mathbf{w}_t), D_t h_t\rangle$$

$$\quad - \langle\nabla P(\mathbf{w}_t), (D_{t+1}-D_t)h_{t+1}\rangle$$

$$= I_1^t + I_2^t + I_3^t$$

Let us first bound $I_1^t$,

$$\mathbb{E}_t[I_1^t] \overset{(a)}{=} -(1-\eta_1)\langle\nabla P(\mathbf{w}_t), \mathbb{E}_t[D_t\nabla g_{i_t}(\mathbf{w}_t; \xi)^\top\nabla f(\mathbf{u}_{i_t})]\rangle$$

$$= -(1-\eta_1)\langle\nabla P(\mathbf{w}_t), \mathbb{E}_t[D_t\nabla g_{i_t}(\mathbf{w}_t; \xi)^\top\nabla f(g_{i_t}(\mathbf{w}_t))]\rangle$$

$$\quad + (1-\eta_1)\langle\nabla P(\mathbf{w}_t), \mathbb{E}_t[D_t\nabla g_{i_t}(\mathbf{w}_t; \xi)^\top(\nabla f(\mathbf{u}_{i_t}) - \nabla f(g_{i_t}(\mathbf{w}_t)))]\rangle$$

$$\le -(1-\eta_1)\|\nabla P(\mathbf{w}_t)\|_{D_t}^2$$

$$\quad + (1-\eta_1)\|D_t^{-1/2}\nabla P(\mathbf{w}_t)\|\mathbb{E}_t[\|D_t^{-1/2}\nabla g_{i_t}(\mathbf{w}_t; \xi)^\top(\nabla f(\mathbf{u}_{i_t}) - \nabla f(g_{i_t}(\mathbf{w}_t)))\|]$$

$$\overset{(b)}{\le} -(1-\eta_1)\|\nabla P(\mathbf{w}_t)\|_{D_t}^2 + \frac{(1-\eta_1)\|\nabla P(\mathbf{w}_t)\|_{D_t}^2}{2}$$

$$\quad + \frac{(1-\eta_1)\mathbb{E}_t[\|D_t^{-1/2}\nabla g_{i_t}(\mathbf{w}_t; \xi)^\top(\nabla f(\mathbf{u}_{i_t}) - \nabla f(g_{i_t}(\mathbf{w}_t)))\|^2]}{2}$$

$$\le -\frac{(1-\eta_1)}{2}\|\nabla P(\mathbf{w}_t)\|_{D_t}^2 + \frac{(1-\eta_1)}{2}\mathbb{E}_t[\|\nabla g_{i_t}(\mathbf{w}_t; \xi)^\top(\nabla f(\mathbf{u}_{i_t}) - \nabla f(g_{i_t}(\mathbf{w}_t))\|_{D_t}^2]$$

$$\overset{(c)}{\le} -\frac{(1-\eta_1)}{2}(\epsilon + C_g^2 C_f^2)^{-1/2}\|\nabla P(\mathbf{w}_t)\|^2 + \frac{1}{2}\epsilon^{-1/2}C_g^2 L_f^2\mathbb{E}[\|g_{i_t}(\mathbf{w}_t) - \mathbf{u}_{i_t}\|^2]$$

$$\tag{23}$$

where equality $(a)$ is due to $\nabla P(\mathbf{w}_t) = \mathbb{E}_{i_t,\xi}[\nabla g_{i_t}(\mathbf{w}_t;\xi)^\top \nabla f(g_{i_t}(\mathbf{w}_t))]$, where $i_t$ and $\xi$ are independent. The inequality $(b)$ is according to $ab \le a^2/2 + b^2/2$. The last inequality $(c)$ is due to $\epsilon^{-1/2}\mathbf{I} \ge \|D_t\mathbf{I}\| = \|\frac{1}{\sqrt{\epsilon\mathbf{I}+\hat{v}_{t+1}}}\| \ge \|(\epsilon\mathbf{I}+C_g^2C_f^2)^{-1/2}\| = (\epsilon+C_g^2C_f^2)^{-1/2}\mathbf{I}$, $(1-\eta_1) \le 1$ and

$$
\begin{aligned}
\mathbb{E}_t[\|\nabla g_{i_t}(\mathbf{w}_t;\xi)^\top(\nabla f(\mathbf{u}_{i_t}) - \nabla f(g_{i_t}(\mathbf{w}_t)))\|_{D_t}^2] \\
\le \epsilon^{-1/2}C_g^2\mathbb{E}_t[\|\nabla f(\mathbf{u}_{i_t}) - \nabla f(g_{i_t}(\mathbf{w}_t))\|_{\mathbf{I}}^2] \\
\le \epsilon^{-1/2}C_g^2 L_f^2\mathbb{E}_t[\|g_{i_t}(\mathbf{w}_t) - \mathbf{u}_{i_t}\|^2]
\end{aligned}
\tag{24}
$$

For $I_2^t$ and $I_3^t$, we have

$$
\begin{aligned}
\mathbb{E}_t[I_2^t] &= -\eta_1\langle\nabla P(\mathbf{w}_t) - \nabla P(\mathbf{w}_{t-1}), D_t h_t\rangle - \eta_1\langle\nabla P(\mathbf{w}_{t-1}), D_t h_t\rangle \\
&\le \eta_1 L\alpha^{-1}\|\mathbf{w}_t - \mathbf{w}_{t-1}\|^2 - \eta_1\langle\nabla P(\mathbf{w}_{t-1}), D_t h_t\rangle \\
&= \eta_1 L\alpha^{-1}\|\mathbf{w}_t - \mathbf{w}_{t-1}\|^2 + \eta_1(I_1^{t-1} + I_2^{t-1} + I_3^{t-1}) \\
&\le \eta_1 L\alpha d(1-\eta_2)^{-1}(1-\tau)^{-1} + \eta_1(I_1^{t-1} + I_2^{t-1} + I_3^{t-1})
\end{aligned}
\tag{25}
$$

where the last equation applies Lemma 7.

$$
\begin{aligned}
\mathbb{E}_t[I_3^t] &= -\langle\nabla P(\mathbf{w}_t), (D_{t+1} - D_t)h_{t+1}\rangle = -\sum_{i'=1}^d \nabla_{i'} P(\mathbf{w}_t)((\epsilon+\hat{v}_t^{i'})^{-1/2} - (\epsilon+\hat{v}_{t+1}^{i'})^{-1/2})h_{t+1}^{i'} \\
&\le \|\nabla P(\mathbf{w}_t)\|\|h_{t+1}\|\sum_{i'=1}^d((\epsilon+\hat{v}_t^{i'})^{-1/2} - (\epsilon+\hat{v}_{t+1}^{i'})^{-1/2}) \\
&\le C_g^2 C_f^2\sum_{i'=1}^d((\epsilon+\hat{v}_t^{i'})^{-1/2} - (\epsilon+\hat{v}_{t+1}^{i'})^{-1/2})
\end{aligned}
\tag{26}
$$

By combining Equation (24), (25) and (26) together,

$$
\begin{aligned}
\mathbb{E}_t[I_1^t + I_2^t + I_3^t] &\le -\frac{(1-\eta_1)}{2}(\epsilon+C_g^2C_f^2)^{-1/2}\|\nabla P(\mathbf{w}_t)\|^2 + \frac{1}{2}\epsilon^{-1/2}C_g^2 L_f^2\mathbb{E}_t[\|g_{i_t}(\mathbf{w}_t) - \mathbf{u}_{i_t}\|^2] \\
&\quad + \eta_1 L\alpha d(1-\eta_2)^{-1}(1-\tau)^{-1} + \eta_1(I_1^{t-1} + I_2^{t-1} + I_3^{t-1}) \\
&\quad + C_g^2 C_f^2\sum_{i'=1}^d((\epsilon+\hat{v}_t^{i'})^{-1/2} - (\epsilon+\hat{v}_{t+1}^{i'})^{-1/2})
\end{aligned}
\tag{27}
$$

Define the Lyapunov function

$$
\mathcal{L}_t = P(\mathbf{w}_t) - c_t\langle\nabla P(\mathbf{w}_{t-1}), D_t h_t\rangle
\tag{28}
$$

where $c_t$ and $c$ will be defined later.

$$
\begin{aligned}
&\mathbb{E}_t[\mathcal{L}_{t+1} - \mathcal{L}_t] \\
&= P(\mathbf{w}_{t+1}) - P(\mathbf{w}_t) - c_{t+1}\langle\nabla P(\mathbf{w}_t), D_{t+1}h_{t+1}\rangle + c_t\langle\nabla P(\mathbf{w}_{t-1}), D_t h_t\rangle \\
&\le -(c_{t+1}+\alpha)\langle\nabla P(\mathbf{w}_t), D_{t+1}h_{t+1}\rangle + \frac{L}{2}\|\mathbf{w}_{t+1} - \mathbf{w}_t\|^2 + c_t\langle\nabla P(\mathbf{w}_{t-1}), D_t h_t\rangle \\
&= (c_{t+1}+\alpha)(I_1^t + I_2^t + I_3^t) + \frac{L}{2}\|\mathbf{w}_{t+1} - \mathbf{w}_t\|^2 - c_t(I_1^{t-1} + I_2^{t-1} + I_3^{t-1}) \\
&\overset{\substack{Eqn\ (27)\ \text{and}\ Lemma\ 7}}{\le} -(\alpha+c_{t+1})\frac{(1-\eta_1)}{2}(\epsilon+C_g^2C_f^2)^{-1/2}\|\nabla P(\mathbf{w}_t)\|^2 \\
&\quad + (\alpha+c_{t+1})\eta_1 L\alpha d(1-\eta_2)^{-1}(1-\tau)^{-1} + \eta_1(\alpha+c_{t+1})(I_1^{t-1} + I_2^{t-1} + I_3^{t-1}) \\
&\quad + (\alpha+c_{t+1})C_g^2 C_f^2\sum_{i'=1}^d((\epsilon+\hat{v}_{i'}^t)^{-1/2} - (\epsilon+\hat{v}_{i'}^{t+1})^{-1/2}) \\
&\quad + \frac{L}{2}\alpha^2 d(1-\eta_2)^{-1}(1-\tau)^{-1} - c_t(I_1^{t-1} + I_2^{t-1} + I_3^{t-1}) + \frac{\epsilon^{-1/2}C_g^2 L_f^2(\alpha+c_{t+1})}{2}\|g_{i_t}(\mathbf{w}_t) - \mathbf{u}_{i_t}\|^2
\end{aligned}
\tag{29}
$$

By setting $\alpha_{t+1} \le \alpha_t = \alpha$, $c_t = \sum_{p=t}^{\infty}(\prod_{j=t}^{p}\eta_1)\alpha_j$, and $c = (1 + (1-\eta_1)^{-1})\epsilon^{-\frac{1}{2}}C_g^2 L_f^2$, we have

$$c_t \le (1-\eta_1)^{-1}\alpha_t, \quad \frac{2(\alpha + c_{t+1})}{\alpha}\beta\epsilon^{-1/2}C_g^2 L_f^2 \le c\beta, \quad \eta_1(\alpha + c_{t+1}) = c_t \qquad (30)$$

As a result, $\eta_1(\alpha + c_{t+1})(I_1^{t-1} + I_2^{t-1} + I_3^{t-1}) - c_t(I_1^{t-1} + I_2^{t-1} + I_3^{t-1}) = 0$

$$\mathbb{E}_t[\mathcal{L}_{t+1} - \mathcal{L}_t] \le -(\alpha + c_{t+1})\frac{(1-\eta_1)}{2}(\epsilon + C_g^2 C_f^2)^{-1/2}\|\nabla P(\mathbf{w}_t)\|^2$$

$$+ (\alpha + c_{t+1})\eta_1 L\alpha d(1-\eta_2)^{-1}(1-\tau)^{-1} + \frac{L}{2}\alpha^2 d(1-\eta_2)^{-1}(1-\tau)^{-1}$$

$$+ (\alpha + c_{t+1})C_g^2 C_f^2 \sum_{i'=1}^{d}((\epsilon + \hat{v}_{i'}^{t})^{-1/2} - (\epsilon + \hat{v}_{i'}^{t+1})^{-1/2})$$

$$+ \frac{(\alpha + c_{t+1})}{2}\epsilon^{-1/2}C_g^2 L_f^2\|g_{i_t}(\mathbf{w}_t) - \mathbf{u}_{i_t}\|^2$$

$$\le -\alpha\frac{(1-\eta_1)}{2}(\epsilon + C_g^2 C_f^2)^{-1/2}\|\nabla P(\mathbf{w}_t)\|^2$$

$$+ 2\eta_1 L\alpha^2 Td(1-\eta_1)^{-1}(1-\eta_2)^{-1}(1-\tau)^{-1} + \frac{L}{2}T\alpha^2 d(1-\eta_2)^{-1}(1-\tau)^{-1}$$

$$+ 2(1-\eta_1)^{-1}\alpha C_g^2 C_f^2 \sum_{i'=1}^{d}((\epsilon + \hat{v}_{i'}^{t})^{-1/2} - (\epsilon + \hat{v}_{i'}^{t+1})^{-1/2}) + \frac{c\alpha}{4}\sum_{t=1}^{T}\mathbb{E}_t[\|g_{i_t}(\mathbf{w}_t) - \mathbf{u}_{i_t}\|^2]$$

$$(31)$$

where the last inequality is due to equation (30) such that we have $2(\alpha + c_{t+1})\epsilon^{-1/2}C_g^2 L_f^2 \le c\alpha$, and $\alpha + c_{t+1} \le 2(1-\eta_1)^{-1}\alpha$.

Then by rearranging terms, and taking summation from $1, \cdots, T$ of equation (31), we have

$$\sum_{t=1}^{T}\alpha\frac{(1-\eta_1)}{2}(\epsilon + C_g^2 C_f^2)^{-1/2}\|\nabla P(\mathbf{w}_t)\|^2 \le \sum_{t=1}^{T}\mathbb{E}_t[\mathcal{L}_t - \mathcal{L}_{t+1}]$$

$$+ 2\eta_1 L\alpha^2 Td(1-\eta_1)^{-1}(1-\eta_2)^{-1}(1-\tau)^{-1} + LT\alpha^2 d(1-\eta_2)^{-1}(1-\tau)^{-1}$$

$$+ 2(1-\eta_1)^{-1}\alpha C_g^2 C_f^2 \sum_{t=1}^{T}\sum_{i'=1}^{d}((\epsilon + \hat{v}_{i'}^{t})^{-1/2} - (\epsilon + \hat{v}_{i'}^{t+1})^{-1/2}) + c\alpha\sum_{t=1}^{T}\mathbb{E}_t[\|g_{i_t}(\mathbf{w}_t) - \mathbf{u}_{i_t}\|^2]$$

$$\le \mathbb{E}[\mathcal{L}_1] - \mathbb{E}[\mathcal{L}_{T+1}]$$

$$+ 2\eta_1 L\alpha^2 Td(1-\eta_1)^{-1}(1-\eta_2)^{-1}(1-\tau)^{-1} + LT\alpha^2 d(1-\eta_2)^{-1}(1-\tau)^{-1}$$

$$+ 2(1-\eta_1)^{-1}\alpha C_g^2 C_f^2 \sum_{i'=1}^{d}((\epsilon + \hat{v}_0^{i'})^{-1/2}) + c\alpha\sum_{t=1}^{T}\mathbb{E}_t[\|g_{i_t}(\mathbf{w}_t) - \mathbf{u}_{i_t}\|^2]$$

$$(32)$$

By combing with Lemma 3, We finish the proof. $\qquad\square$