# OpenReview forum: "Stochastic Optimization of Areas Under Precision-Recall Curves with Provable Convergence"
_NeurIPS.cc/2021/Conference — NeurIPS 2021 Poster_

### Official Review · Reviewer_Ng3u · 2021-07-10

**Rating:** 7
**Confidence:** 3

**Summary:**

This paper studies classification in an imbalanced data setting, where the goal is to obtain large area under the precision-recall curve (AUPRC). This paper proposes a stochastic optimization method procedure for directly optimizing the AUPRC. Their procedure is based on optimizing a continuous approximation to the average precision (AP), which is obtained by substituting non-continuous indicator functions with continuous loss surrogates, and directly optimizing the resulting objective.

The optimization algorithm proposed by this paper is called SOAP, and it is essentially a first order optimization method (e.g. SGD or Adam) with a better estimate for the stochastic gradient of the objective. The key for the improved stochastic estimate is to control the variance by tracking an exponential moving average to compute terms in the denominator of the gradient. SOAP has convergence guarantees that the average gradient norm of the surrogate AP objective will go to zero.

Empirically, the authors apply SOAP to a variety of settings with imbalanced datasets such as imbalanced CIFAR, Melanoma datasets in the image classification settings, molecular prediction datasets with graph NNs, and drug discovery datasets with graph NNs. They show that SOAP can outperform competing baselines, particularly in the setting where the dataset is extremely imbalanced.

**Ethical Concerns:**

N/A.

**Limitations And Societal Impact:**

Yes.

**Main Review:**

This paper has nice theoretical and empirical contributions for the setting with very imbalanced data, I recommend accept.

Originality: the paper proposes novel methods and analysis.

Quality: both theoretical and empirical claims are well-supported in this paper. The experiments are very thorough and conducted for a lot of datasets for a wide range of architectures and settings.

Clarity: The submission is clearly written and easy to follow.

Significance: the paper proposes a clean framework for optimizing AUPRC directly with theoretical guarantees and good empirical performance even in the setting with small minibatch updates, which allows the method to scale to deep learning. This is a nice contribution over prior work.

A question/concern for the authors:

-- One limitation of the loss surrogate seems to be that, depending on the exact surrogate loss $\ell$, it no longer seems to be an upper bound for the AP. In contrast, in typical settings where loss surrogates are used, the loss surrogate tends to be an upper bound (perhaps up to some constant factor scaling) on the target loss (e.g. standard classification where 0/1 loss is the target, and cross entropy is the surrogate). Can the authors comment on which choice of loss surrogate is used in the paper, and how the the value of the loss surrogate will generally compare to the true AP?

=====================================================================

Update after rebuttal: Thanks to the authors for the responses. I have read the other reviews and responses and will keep my score the same.

**Time Spent Reviewing:**

3

---

> ### Author Response · Authors · 2021-08-10
> **Thank you for your positive comments!**
>
> Thank you for your positive rating!
>
> **Q1**: Can the authors comment on which choice of loss surrogate is used in the paper, and how the value of the loss surrogate will generally compare to the true AP?
>
> **Response**: We use the squared hinge loss for our method in the paper (cf. line 137). We have conducted an experiment to check the consistency between the convergence of true AP vs the surrogate objective (-$P(\mathbf w)$) that is implied by the loss surrogate by comparing the curve of true AP vs iteration and the curve of surrogate objective vs iteration on the training data. We report the AP and the surrogate objective on the training data at 10 different iterations in the training process over 5 independent runs. The results are shown in the following table, which demonstrates that optimizing the surrogate loss is consistent with optimizing the original metric. We will include more discussion and such results in the revision.
>
> |     CIFAR10    |        Iters      |           0          |         3.2e3        |         6.4e3        |         9.6e3       |        1.28e4       |         1.6e4       |        1.92e4       |
> |:--------------:|:-----------------:|:--------------------:|:--------------------:|:--------------------:|:-------------------:|:-------------------:|:-------------------:|:-------------------:|
> |        AP      |      mean(std)    |     0.039 (0.040)    |     0.314 (0.174)    |     0.523(0.04 4)    |     0.545(0.025)    |     0.569(0.011)    |     0.576(0.013)    |     0.582(0.008)    |
> |      -P(w)     |     mean (std)    |      0.019(0.003)    |      0.109(0.102)    |      0.276(0.109)    |     0.357(0.028)    |     0.382(0.015)    |     0.391(0.020)    |     0.396(0.008)    |
>
>
> |     CIFAR100    |       Iters      |           0         |         3.2e3       |         6.4e3       |         9.6e3       |        1.28e4       |         1.6e4       |        1.92e4       |
> |:---------------:|:----------------:|:-------------------:|:-------------------:|:-------------------:|:-------------------:|:-------------------:|:-------------------:|:-------------------:|
> |        AP       |     mean(std)    |     0.032(0.033)    |     0.292(0.232)    |     0.338(0.253)    |     0.523(0.106)    |     0.571(0.024)    |     0.580(0.023)    |     0.583(0.029)    |
> |       -P(w)     |     mean(std)    |     0.015(0.004)    |     0.105(0.102)    |     0.215(0.162)    |     0.307(0.145)    |     0.380(0.038)    |     0.390(0.034)    |     0.394(0.038)    |
>
> We also include the python code for plotting the two curves of AP score and Objective vs the number of iterations on CIFAR10 and CIFAR100 in a separate comment below for your reference. You can run the code to better visualize the results.
>
> Thank you!

---

### Official Review · Reviewer_PDbZ · 2021-07-13

**Rating:** 7
**Confidence:** 3

**Summary:**

This paper investigates the imbalanced classification problems under the areas under precision-recall curves (AUPRC). To handle the nondifferentiability of the indicator functions in  the averaged precision (AP) estimators, they propose a differentiable surrogate functions such as squared hinge loss to replace the indicator function, leading to a compositional optimization problem. To solve this compositional problem, the authors use the moving average forward estimators in Wang et al., 2017, to estimate g_{xi}(w) in the gradient estimator of the objective function. For the outer updates, the authors introduce both the SGD-type and Adam-type gradient updates. Theoretically, this paper is the first one to provide the finite-time convergence analysis as well as a computational complexity for stochastic optimization of AUPRC. I am happy to see that the proposed method perform quite well in various imbalanced classification problems. The experiments are clear, comprehensive and involve a variety of datasets like CIFAR10, 100, HIV, MUV, and data for drug discovery, and test different types of neural networks such as ResNet. Under such a setup, they show their method outperforms existing state-of-the-arts via a notable margin (particularly for those highly imbalanced datasets).

**Limitations And Societal Impact:**

Yes

**Main Review:**

========================================================
After rebuttal:

I am satisfied with the good response and additional empirical results. Therefore, I keep my score towards acceptance.

========================================================

This paper investigates the imbalanced classification problems under the areas under precision-recall curves (AUPRC). To handle the nondifferentiability of the indicator functions in  the averaged precision (AP) estimators, they propose a differentiable surrogate functions such as squared hinge loss to replace the indicator function, leading to a compositional optimization problem. To solve this compositional problem, the authors use the moving average forward estimators in Wang et al., 2017, to estimate g_{xi}(w) in the gradient estimator of the objective function. For the outer updates, the authors introduce both the SGD-type and Adam-type gradient updates. Theoretically, this paper is the first one to provide the finite-time convergence analysis as well as a computational complexity for stochastic optimization of AUPRC. I am happy to see that the proposed method perform quite well in various imbalanced classification problems. The experiments are clear, comprehensive and involve a variety of datasets like CIFAR10, 100, HIV, MUV, and data for drug discovery, and test different types of neural networks such as ResNet. Under such a setup, they show their method outperforms existing state-of-the-arts via a notable margin (particularly for those highly imbalanced datasets).

For my own interest, I have several questions as below:

1.	Can the authors explain a little bit why the moving average estimators involve the momentum-type updates, i.e., (1-\gamma)u+\gamma g? Does this help to reduce the variance of the estimators? What happens if such momentum-type updates are removed?
2.	I miss something but do the authors investigate the impact of different surrogate functions on the final empirical performance of SOAP? Can the authors provide a guideline on how to choose the surrogate functions for different problems?
3.	It seems to me that for a relatively larger imbalanced ratio, e.g., 12% and above, as shown in Table 4 in the appendix A, the proposed method is sometimes worse than MinMax method. Can the authors provide a comparison for a higher ratio such as 20%? I am wondering whether the proposed method is comparable to MinMax even for a relatively larger ratio. I am concerned about this because it may happen that the extremely imbalanced datasets are not that common in practice.


Despite the above questions, I still like this paper due to two major reasons. First, the surrogate loss idea is clever although not hard to come up with. This reformulation of the AC really helps to design provable and faster gradient-based algorithms. Furthermore, the compositional optimization view point, the adoption of the moving average estimators and the developed nonasymptotic theoretical analysis are interesting. Second, I am happy to see that the proposed method can work quite well in practice, and the improvements over existing baselines are significant for highly imbalanced datasets. For these reasons, I tend to accept this paper.


**Time Spent Reviewing:**

6

---

> ### Author Response · Authors · 2021-08-10
> **Thank you for your positive comments!**
>
> Thank you for your positive rating!
>
> **Q1**: Can the authors explain a little bit why the moving average estimators involve the momentum-type updates, i.e., $(1-\gamma)\mathbf u+\gamma g$? Does this help to reduce the variance of the estimators? What happens if such momentum-type updates are removed?
>
> **Response**:
> a.	 The moving-average estimator $\mathbf u_{t+1} = (1-\gamma) \mathbf u_t+\gamma g$ is usually referred to the momentum-type updates, e.g., please refer to [ref1, ref2]. This naming is coming from that it is similar to the update used in the conventional momentum methods (e.g., heavy-ball method).
>
> b.	 Yes, it helps reduce the variance of the estimators. In our algorithm, the moving average estimator u_i is used for tracking g_i. The variance reduction property can be seen from Lemma 3. Dividing both sides by T, and setting $\gamma, \alpha$ appropriately, e.g., the values in Theorem 1 (i.e., $\alpha = \frac{1}{n_+^{3/5}T^{3/5}}$, and $\gamma = \frac{n_+^{2/5}}{T^{2/5}}$), the right hand side of (8) dividing by $T$ is diminishing in the order of $O(1/T^{2/5})$. This implies that the averaged variance of $\mathbf u_{i_t}$ compared with $g_{i_t}(\mathbf w_t)$ across all iterations is diminishing.
>
> c.	If the momentum-type updates are removed (equivalently setting $\gamma=1$, the algorithm is not guaranteed to converge with a small mini-batch size. In this case, the gradient is approximated by a naïve method that simply computes the gradient of the AP computed only on the mini-batch data, i.e., the approximated gradient is given by $\frac{1}{B_+}\sum_{\mathbf x_i\in\mathcal B_+}\nabla \widehat g_{\mathbf x_i}(\mathbf w)\nabla f(\widehat g_{\mathbf x_i}(\mathbf w))$, where $\widehat g_{\mathbf x_i}(\mathbf w)$ is the mini-batch version of $g_{\mathbf x_i}(\mathbf w)$. We can show that the optimization error will depend on the mini-batch size used to compute the $\widehat g_{\mathbf x_i}(\mathbf w)$. Hence, the algorithm will have a large optimization error (poor performance) when mini-batch size is small. This has been verified by existing studies [6, 45, 46] (please refer to lines 93-94 as well).
>
> [ref1]: Ashok Cutkosky, Harsh Mehta.  Momentum Improves Normalized SGD. ICML, 2020.
>
> [ref2]: Yanli Liu, Yuan Gao, Wotao Yin. An Improved Analysis of Stochastic Gradient Descent with Momentum. NeurIPS, 2020.
>
> **Q2**:  I miss something but do the authors investigate the impact of different surrogate functions on the final empirical performance of SOAP? Can the authors provide a guideline on how to choose the surrogate functions for different problems?
>
> **Response**: In the experiments, we use the squared hinge loss as the surrogate function for our method.  However, our theory holds as long as the surrogate loss is a smooth function. We have conducted an experiment on CIFAR10, CIFAR100, HIV, MUV datasets by using other two surrogate function, i.e., the logistic loss $\ell(\mathbf w, \mathbf x_s, \mathbf x_i) = - \log \frac{1}{1+\exp(-c(h_{\mathbf w}(\mathbf x_i) – h_{\mathbf w}(\mathbf x_s)))}$ and the sigmoid loss $\frac{1}{1+\exp(c(h_{\mathbf w}(\mathbf x_i) – h_{\mathbf w}(\mathbf x_s)))}$, where $c$ is a hyper-parameter.  The results are shown in the following table. We can see that the results of using different surrogate losses are comparable, and better than baselines. It is difficult to provide a guideline what is the best surrogate functions for different problems. It is usually problem-dependent.
>
>
> |                 |     Squared Hinge   Loss(ResNet18)    |     Logistic Loss   (ResNet18)    |     Sigmoid Loss   (ResNet18)    |
> |-----------------|:-------------------------------------:|:---------------------------------:|:--------------------------------:|
> |      CIFAR10    |             0.7644 (± 3e-3)           |           0.7690 (± 1e-3)         |          0.7653 (± 3e-3)         |
> |     CIFAR100    |             0.6237 (± 2e-3)           |           0.6235 (± 2e-3)         |           0.6380(± 3e-3)         |
>
>
>
> |            |     Squared Hinge Loss (MPNN)    |     Logistic Loss  （MPNN）    |     Sigmoid Loss     (MPNN)    |
> |:----------:|:--------------------------------:|:------------------------------:|:------------------------------:|
> |     HIV    |          0.3401 (± 4e-3)         |         0.3617 (± 3e-3)        |         0.3629 (± 6e-3)        |
> |     MUV    |          0.3365 (± 8e-4)         |         0.3356 (± 2e-4)        |         0.3362 (± 9e-4)        |
>
>
> **Q3**: It seems to me that for a relatively larger imbalanced ratio, e.g., 12\% and above, as shown in Table 4 in the appendix A, the proposed method is sometimes worse than the MinMax method. Can the authors provide a comparison for a higher ratio such as 20\%? I am wondering whether the proposed method is comparable to MinMax even for a relatively larger ratio. I am concerned about this because it may happen that the extremely imbalanced datasets are not that common in practice.
>
> **Response**: We have conducted an experiment on CIFAR10 and CIFAR100 datasets by constructing their imbalanced version with a large imbalance ratio of 20\%. The results of our method and MinMax for learning ResNet18 are shown in the following Table. We can see our methods consistently perform better than the MinMax.
>
> |     Imbalance   Ratio = 20\%    |     SOAP (ResNet18)    |     MinMax (ResNet18)    |
> |:-------------------------------:|:----------------------:|:------------------------:|
> |              CIFAR10            |     0.8349 (± 1e-3)    |      0.8216 (± 1e-3)     |
> |             CIFAR100            |     0.6818 (± 1e-3)    |      0.6370 (± 6e-3)     |
>
> Thank you!

---

### Official Review · Reviewer_WXNT · 2021-07-16

**Rating:** 6
**Confidence:** 4

**Summary:**

The author presented a technique for optimizing the Area under the Precision-Recall curve (AUCPR). The proposed technique works by replacing the indicator function in the empirical AUCPR formulation with a smooth loss function and then decompose the formula as a sum of compositional functions. The author then proposed a learning algorithm based on stochastic compositional optimization. Finally, the author shows the effectiveness of the proposed model on various classification tasks.

**Limitations And Societal Impact:**

Yes

**Main Review:**

**Strengths**

1. Optimizing AUCPR is useful, particularly in the setting of imbalanced datasets. Having a method that works in deep learning settings is useful for many applications.
2. The use of the stochastic compositional optimization technique in the proposed algorithm is interesting.
3. The author provides a convergence analysis of the model.
4. The experiments show the benefit of the model, and the model is less sensitive to batch size.

**Weaknesses, Concerns, and Questions**

1. The idea of replacing the indicator functions in the fractional metrics with smooth surrogate functions and directly perform optimization on that objective using neural networks is not new. Multiple related papers have tried that on several different metrics. The new idea in this paper is to use the stochastic compositional optimization technique to solve the resulting objective. I am not sure if this contribution only can warrant a NeurIPS acceptance.
2. One of the challenges in optimizing a non-decomposable metric is that calculating the gradients of the surrogate requires computations over all samples. The author proposed a method to approximate the gradient using stochastic samples. How does this approximation affect the performance of the model?
3. One of the disadvantages of constructing a learning objective on a fractional metric by putting a smooth (convex) surrogate on both numerator and denominator is that the value of this fraction of surrogates may deviate a lot from the value of the original metric due to fractional computation. This is in contrast with a simple surrogate on the accuracy metric, e.g., hinge loss which has a monotonic relationship with the zero-one loss metric. Has the author investigated this potential misalignment of the surrogate vs. the original metric?

**Time Spent Reviewing:**

4

---

> ### Author Response · Authors · 2021-08-10
> **Thank you for your positive comments!**
>
> Thank you for agreeing that our algorithm is interesting and useful. We believe your concerns and questions can be easily addressed by our responses below. We hope the reviewer can increase your score if we address your concerns.
>
>
> **Q1**: Is the contribution enough to warrant a NeurIPS acceptance.
>
> **Response**:  We would like to summarize our contributions again. **Theoretical Contribution**: our stochastic algorithm is **the first one** that optimizes Average Precision with a theoretical convergence guarantee. We cast the problem into a new family of finite-sum of stochastic compositional functions. Hence, our algorithms can be potentially used for solving many other problems belonging to this family. **Empirical Contribution**: we have conducted extensive experiments of the proposed algorithms for deep learning with imbalanced data that cover image classification on 3 image datasets (CIFAR10, CIFAR100, Melanoma) and graph classification on 5 graph datasets (HIV, MUV, MIT AICURES, ToxCast, Tox21). Our studies not only verify the theory but also demonstrate the effectiveness of the proposed method with dramatic improvements on baselines, e.g., 30% improvement on MUV data with 0.2% positive examples. We believe these are significant contributions that would benefit the community working on imbalance data.
>
>
> **Q2**:  The author proposed a method to approximate the gradient using stochastic samples. How does this approximation affect the performance of the model?
>
> **Response**: Although the gradient is approximated at each iteration, overall the proposed algorithm is guaranteed to converge to a first-order optimal solution of the non-convex objective. Our method for approximating the gradient using stochastic samples is innovatively designed such that the algorithm has rigorous theoretical convergence guarantee (Theorem 1 and Theorem 2).  In contrast, a naïve approximation method simply by using the mini-batch to compute a surrogate AP score and its gradient provides no convergence guarantee. Such a simple method has been used by many previous works including the two baselines SmoothAP [6] and FastAP [4], and existing studies e.g., [6, 45, 46] have observed that this simple method has poor performance when the mini-batch size is small.  Hence, our approximation method by using the moving average estimators – the $\mathbf u$ sequences,  is better than the previous approximation method, which is not only supported by our theory but also evidenced by experimental results: (i) our method performs better than SmoothAP and FastAP; (ii) our method is insensitive to the mini-batch size consistent with our theory (Figure 1 right).
>
>
> **Q3**:  Has the author investigated this potential misalignment of the surrogate vs. the original metric to the fractional objective?
>
> **Response**: This is a nice point. We empirically checked the consistency between the convergence of true AP vs the surrogate objective (-$P(\mathbf w)$)  by comparing the curve of true AP vs iteration and the curve of surrogate objective vs iteration on the training data. The results shown in the following table demonstrate on CIFAR10 and CIFAR100 data that optimizing the surrogate loss is consistent with optimizing the original metric. We will include more discussion and such results in the revision.
>
>
> |     CIFAR10    |        Iters      |           0          |         3.2e3        |         6.4e3        |         9.6e3       |        1.28e4       |         1.6e4       |        1.92e4       |
> |:--------------:|:-----------------:|:--------------------:|:--------------------:|:--------------------:|:-------------------:|:-------------------:|:-------------------:|:-------------------:|
> |        AP      |      mean(std)    |     0.039 (0.040)    |     0.314 (0.174)    |     0.523(0.04 4)    |     0.545(0.025)    |     0.569(0.011)    |     0.576(0.013)    |     0.582(0.008)    |
> |      -P(w)     |     mean (std)    |      0.019(0.003)    |      0.109(0.102)    |      0.276(0.109)    |     0.357(0.028)    |     0.382(0.015)    |     0.391(0.020)    |     0.396(0.008)    |
>
>
> |     CIFAR100    |       Iters      |           0         |         3.2e3       |         6.4e3       |         9.6e3       |        1.28e4       |         1.6e4       |        1.92e4       |
> |:---------------:|:----------------:|:-------------------:|:-------------------:|:-------------------:|:-------------------:|:-------------------:|:-------------------:|:-------------------:|
> |        AP       |     mean(std)    |     0.032(0.033)    |     0.292(0.232)    |     0.338(0.253)    |     0.523(0.106)    |     0.571(0.024)    |     0.580(0.023)    |     0.583(0.029)    |
> |       -P(w)     |     mean(std)    |     0.015(0.004)    |     0.105(0.102)    |     0.215(0.162)    |     0.307(0.145)    |     0.380(0.038)    |     0.390(0.034)    |     0.394(0.038)    |
>
>
> We also include the python code for plotting the two curves of AP score and Objective vs the number of iterations on CIFAR10 and CIFAR100 in a separate comment for your reference. You can run the code to better visualize the results.
>
> Thank you!

---

> > ### Comment · Reviewer_WXNT · 2021-08-22
> > **Thanks for the response**
> >
> > Thanks the authors for addressing my questions and concerns.
> > I have read the detailed responses as well as other reviewer's comments.
> > I mostly satisfy with the author responses. Therefore I have increased my score.

---

> > > ### Author Response · Authors · 2021-08-22
> > > **Thank you for increasing the score!**
> > >
> > > We are glad to hear that our response has addressed your concern. Thank you!

---

> ### Author Response · Authors · 2021-08-21
> **To Reviewer WXNT: Are your concerns addressed? Thank you!**
>
> Dear Reviewer WXNT,
>
> We believe  our response has addressed your concerns. Please let us know if there is anything further that we can clarify.  Given your overall positive comment, we hope that you can upgrade your score if we have addressed your concerns.
>
> Authors

---

### Author Response · Authors · 2021-08-10
**Summary of Comments and Responses.**

First, we thank all reviewers for the time and efforts to evaluate our paper. We thank you for appreciating the merits of our work, including but not limited to:

**1**  “Having a method that works in deep learning settings is useful for many applications.”
“The use of the stochastic compositional optimization technique in the proposed algorithm is interesting.” (Reviewer WXNT)

**2**  “the proposed method perform quite well in various imbalanced classification problems.”   “ Theoretically, this paper is the first one to provide the finite-time convergence analysis as well as a computational complexity for stochastic optimization of AUPRC.”  “The experiments are clear, comprehensive and involve a variety of datasets like CIFAR10, 100, HIV, MUV, and data for drug discovery, and test different types of neural networks such as ResNet.” (Reviewer PDbZ)

**3** “the paper proposes novel methods and analysis.”, “both theoretical and empirical claims are well-supported in this paper.” (Reviewer Ng3u)

While some questions regarding some experimental comparison were raised by the reviewers, we believe that our responses have addressed them.  We are looking forward to further discussions and more than happy to answer any other questions.

Best

Authors

---

### Author Response · Authors · 2021-08-10
**Plots and Python Code for Plotting AP and Surrogate Objective vs Iterations**

The plots showing consistency between AP and Surrogate Objective vs Iterations can be found in the link below for better visualization https://anonymous.4open.science/r/auprc_rebuttal-EE7F/

The Python code for plotting is given below.

import matplotlib.pyplot as plt

import numpy as np

iterations = np.array([ 400,   800,  1200,  1600,  2000,  2400,  2800,  3200,
        3600,  4000,  4400,  4800,  5200,  5600,  6000,  6400,  6800,
        7200,  7600,  8000,  8400,  8800,  9200,  9600, 10000, 10400,
       10800, 11200, 11600, 12000, 12400, 12800, 13200, 13600, 14000,
       14400, 14800, 15200, 15600, 16000, 16400, 16800, 17200, 17600,
       18000, 18400, 18800, 19200, 19600, 20000, 20400, 20800, 21200,
       21600, 22000, 22400, 22800, 23200, 23600, 24000, 24400, 24800,
       25200, 25600])


cifar10_res = np.array([[0.039, 0.057, 0.091, 0.114, 0.152, 0.194, 0.247, 0.279, 0.314, 0.332, 0.369, 0.408, 0.443, 0.458, 0.472, 0.498, 0.523, 0.524, 0.533, 0.54, 0.537, 0.547, 0.55, 0.551, 0.546, 0.546, 0.565, 0.554, 0.572, 0.557, 0.575, 0.565, 0.569, 0.58, 0.567, 0.575, 0.577, 0.579, 0.589, 0.578, 0.576, 0.579, 0.582, 0.578, 0.582, 0.575, 0.578, 0.576, 0.582, 0.578, 0.584, 0.581, 0.57, 0.578, 0.584, 0.583, 0.592, 0.583, 0.586, 0.586, 0.582, 0.583, 0.6, 0.584],
                        [0.04, 0.065, 0.11, 0.129, 0.144, 0.148, 0.163, 0.17, 0.174, 0.168, 0.167, 0.155, 0.128, 0.09, 0.072, 0.065, 0.044, 0.036, 0.028, 0.023, 0.021, 0.018, 0.027, 0.017, 0.025, 0.008, 0.015, 0.011, 0.014, 0.004, 0.015, 0.02, 0.011, 0.008, 0.011, 0.016, 0.006, 0.013, 0.02, 0.011, 0.013, 0.022, 0.013, 0.013, 0.01, 0.014, 0.013, 0.008, 0.008, 0.006, 0.015, 0.01, 0.01, 0.015, 0.013, 0.008, 0.008, 0.014, 0.016, 0.02, 0.014, 0.019, 0.017, 0.017],
                        [0.018, 0.019, 0.02, 0.022, 0.026, 0.036, 0.06, 0.085, 0.109, 0.129, 0.158, 0.177, 0.197, 0.216, 0.234, 0.256, 0.276, 0.298, 0.317, 0.333, 0.336, 0.349, 0.353, 0.355, 0.357, 0.355, 0.377, 0.367, 0.385, 0.369, 0.39, 0.38, 0.382, 0.394, 0.379, 0.385, 0.386, 0.391, 0.401, 0.392, 0.391, 0.393, 0.393, 0.392, 0.392, 0.387, 0.391, 0.388, 0.396, 0.395, 0.398, 0.393, 0.387, 0.392, 0.399, 0.394, 0.404, 0.395, 0.403, 0.394, 0.397, 0.397, 0.411, 0.393],
                        [0.003, 0.004, 0.005, 0.006, 0.01, 0.026, 0.063, 0.082, 0.102, 0.11, 0.121, 0.122, 0.127, 0.113, 0.113, 0.115, 0.109, 0.072, 0.051, 0.038, 0.037, 0.027, 0.035, 0.024, 0.028, 0.015, 0.023, 0.008, 0.015, 0.006, 0.012, 0.024, 0.015, 0.008, 0.018, 0.02, 0.009, 0.012, 0.021, 0.009, 0.02, 0.025, 0.009, 0.013, 0.011, 0.014, 0.013, 0.01, 0.008, 0.004, 0.014, 0.015, 0.01, 0.014, 0.017, 0.009, 0.01, 0.016, 0.015, 0.02, 0.016, 0.02, 0.021, 0.015]]
)

cifar100_res = np.array([[0.032, 0.059, 0.092, 0.12, 0.136, 0.167, 0.23, 0.289, 0.293, 0.311, 0.308, 0.316, 0.33, 0.328, 0.335, 0.344, 0.338, 0.351, 0.363, 0.427, 0.442, 0.456, 0.473, 0.504, 0.523, 0.521, 0.546, 0.547, 0.561, 0.569, 0.556, 0.562, 0.571, 0.574, 0.574, 0.581, 0.567, 0.58, 0.577, 0.574, 0.58, 0.584, 0.585, 0.583, 0.582, 0.58, 0.576, 0.585, 0.584, 0.58, 0.579, 0.582, 0.586, 0.59, 0.593, 0.592, 0.578, 0.584, 0.587, 0.589, 0.583, 0.589, 0.596, 0.594],
                         [0.034, 0.082, 0.129, 0.157, 0.169, 0.178, 0.189, 0.228, 0.232, 0.246, 0.243, 0.249, 0.258, 0.255, 0.258, 0.263, 0.253, 0.257, 0.237, 0.218, 0.204, 0.201, 0.17, 0.134, 0.106, 0.08, 0.069, 0.059, 0.052, 0.037, 0.03, 0.031, 0.024, 0.03, 0.022, 0.034, 0.029, 0.037, 0.025, 0.036, 0.023, 0.03, 0.027, 0.028, 0.021, 0.026, 0.028, 0.039, 0.029, 0.031, 0.03, 0.026, 0.027, 0.041, 0.028, 0.038, 0.031, 0.021, 0.035, 0.027, 0.026, 0.019, 0.021, 0.036],
                         [0.015, 0.016, 0.017, 0.019, 0.024, 0.046, 0.061, 0.077, 0.106, 0.15, 0.17, 0.184, 0.201, 0.204, 0.208, 0.222, 0.215, 0.227, 0.226, 0.244, 0.247, 0.282, 0.291, 0.302, 0.307, 0.302, 0.32, 0.331, 0.36, 0.371, 0.365, 0.371, 0.38, 0.383, 0.385, 0.389, 0.376, 0.388, 0.387, 0.385, 0.39, 0.395, 0.392, 0.395, 0.396, 0.391, 0.386, 0.397, 0.395, 0.392, 0.391, 0.392, 0.397, 0.405, 0.406, 0.409, 0.388, 0.398, 0.4, 0.399, 0.4, 0.404, 0.407, 0.409],
                         [0.004, 0.004, 0.005, 0.007, 0.015, 0.054, 0.079, 0.093, 0.102, 0.121, 0.131, 0.141, 0.153, 0.154, 0.157, 0.169, 0.162, 0.17, 0.168, 0.179, 0.157, 0.15, 0.147, 0.147, 0.145, 0.139, 0.138, 0.109, 0.082, 0.06, 0.048, 0.041, 0.038, 0.044, 0.037, 0.044, 0.038, 0.045, 0.036, 0.044, 0.034, 0.041, 0.037, 0.037, 0.031, 0.035, 0.034, 0.05, 0.038, 0.04, 0.039, 0.034, 0.037, 0.051, 0.04, 0.047, 0.036, 0.033, 0.042, 0.034, 0.033, 0.024, 0.032, 0.041]]
)

plt.figure()

plt.plot(iterations, cifar10_res[0], linewidth = 2, label = 'AP', color = 'green')

plt.fill_between(iterations, cifar10_res[0] + cifar10_res[1],  cifar10_res[0] -cifar10_res[1], alpha = 0.3, color = 'green')

plt.plot(iterations, cifar10_res[2], linewidth = 2, label = 'Objective (-P(w))', color = 'orange')

plt.fill_between(iterations, cifar10_res[2] + cifar10_res[3], cifar10_res[2] - cifar10_res[3], alpha = 0.3, color = 'orange')

plt.legend(fontsize = 15)

plt.xlabel('Iterations', fontsize = 15)

plt.title('CIFAR10 with ResNet18', fontsize = 15)

plt.show()


plt.figure()

plt.plot(iterations, cifar100_res[0], linewidth = 2, label = 'AP', color = 'green')

plt.fill_between(iterations, cifar100_res[0]+cifar100_res[1],  cifar100_res[0] -cifar100_res[1], alpha = 0.3, color = 'green')

plt.plot(iterations, cifar100_res[2], linewidth = 2, label = 'Objective (-P(w))', color = 'orange')

plt.fill_between(iterations, cifar100_res[2] + cifar100_res[3], cifar100_res[2] - cifar100_res[3],  alpha = 0.3, color = 'orange')

plt.legend(fontsize = 15)

plt.xlabel('Iterations',fontsize = 15)

plt.title('CIFAR100 with ResNet18', fontsize = 15)

plt.show()

---

### Decision · Program_Chairs · 2021-09-27

**Decision:**

Accept (Poster)

**Comment:**

The paper proposes a stochastic optimization technique for maximizing the AUPRC metric, a popular metric in the evaluation of models on class-imbalanced problems. The paper makes compelling theoretical contributions, which are then supported by strong experimental results. All the reviewers lean towards accepting the paper.

Some questions were raised about how closely the proposed surrogate approximations align with the true metric, and how the method fares with different choices of surrogates, to which the authors have provided a satisfactory response with additional experimental results. We strongly encourage the authors to include the additional results in the final paper (perhaps in the appendix if there isn't enough space in the main text).